# Huntington's disease-associated ankyrin repeat palmitoyl transferases are rate-limiting factors in lysosome formation and fusion

Győző Szenci[1], Attila Boda[1,2], Anikó Nagy[1,2], Dorottya Károlyi[1,3], András Rubics[1,3], Zsombor Szőke[1], Gergő Falcsik[3,4], Tibor Kovács[4], Péter Lőrincz[1,2], Gábor Juhász[1,5], Szabolcs Takáts[1]*

1 Department of Anatomy, Cell- and Developmental Biology, Eötvös Loránd University, Budapest, Hungary, 2 HAS-ELTE Momentum Vesicular Transport Research Group, Hungarian Academy of Sciences, Budapest, Hungary, 3 Doctoral School of Biology, Eötvös Loránd University, Budapest, Hungary, 4 Department of Genetics, Eötvös Loránd University, Budapest, Hungary, 5 HUN-REN Biological Research Centre Szeged, Institute of Genetics, Szeged, Hungary

* sz.takats@ttk.elte.hu

## Abstract

Protein palmitoylation in the Golgi apparatus is critical for the appropriate sorting of various proteins belonging to secretory and lysosomal systems, and defective palmitoylation can lead to the onset of severe pathologies. HIP14 and HIP14L ankyrin repeat-containing palmitoyl transferases were linked to the pathogenesis of Huntington's disease, however, how perturbation of these Golgi resident enzymes contributes to neurological disorders is yet to be understood. In this study, we investigated the function of Hip14 and Patsas - the Drosophila orthologs of HIP14 and HIP14L, respectively – to uncover their role in secretory and lysosomal membrane trafficking. Using larval salivary gland, a well-established model of the regulated secretory pathway, we found that these PAT enzymes equally contribute to the proper maturation and crinophagic degradation of glue secretory granules by mediating their fusion with the endo-lysosomal compartment. We also revealed that Patsas and Hip14 are both required for lysosomal acidification and biosynthetic transport of various lysosomal hydrolases, and we demonstrated that the rate of secretory granule-lysosome fusion and subsequent acidification positively correlates with the level of Hip14. Furthermore, Hip14 is also essential for proper lysosome morphology and neuronal function in adult brains. Finally, we found that the over-activation of lysosomal biosynthetic transport and lysosomal fusions by the expression of the constitutively active form of Rab2 could compensate for the lysosomal dysfunction caused by the loss of Patsas or Hip14 both in larval salivary glands and neurons. Therefore, we propose that ankyrin repeat palmitoyl transferases act as rate-limiting factors in lysosomal fusions and provide genetic evidence that defective protein palmitoylation and the subsequent lysosomal dysfunction can contribute to the onset of Huntington's disease-like symptoms.

**Data availability statement:** All relevant data are in the manuscript and its supporting information files.

**Funding:** This study was supported by the National Research, Development and Innovation Office of Hungary (www.nkfih.gov.hu): OTKA FK_142508 for ST, Elvonal KKP129797 and OTKA K146634 to GJ, EKÖP-24-4-I-ELTE-484 for GS, PD142943 for AB, OTKA FK138851 to PL, OTKA PD 143786 for TK, STARTING 150612 for TK, DKOP-23_11 for GF, Magyar Tudományos Akadémia (Hungarian Academy of Sciences): BO/00400/23 for ST, LP2022-13/2022 to PL, LP2023-6 to GJ, Excellence Fund of Eötvös Loránd University (www.elte.hu/eka): EKA_2022/045-P302-1 for ST, EKA 2022/045-P101-2 for PL, EKA_2023/071-P025-1 for TK. The funders had no role in study design, data collection and analysis, decision to publish, or preparation of the manuscript.

**Competing interests:** The authors have declared that no competing interests exist.

## Author summary

Growing body of evidence suggests that decreased activity of HIP14 and HIP14L palmitoyl transferases caused by accumulation of mutant Huntingtin and the subsequent alterations in protein palmitoylation play a critical role in the onset of Huntington's disease (HD). However, which cellular processes are perturbed and eventually lead to the emergence of HD due to impaired palmitoylation is still poorly understood. In our study, we used a *Drosophila* model to uncover the role of Hip14 and Patsas (the fly ortholog of HIP14L) in secretory and lysosomal membrane trafficking. We found that silencing of these transferases in larval salivary glands equally disrupts secretory granule-lysosome fusion, their subsequent acidification, and proper trafficking of lysosomal hydrolases. While overexpression of Hip14 resulted in the acceleration of these processes. We also observed that neuron-specific loss of Hip14 not only perturbs lysosome formation but also results in a progressive decline in neuromuscular functions. Importantly, both lysosomal and neuronal defects emerging in Hip14 and Patsas deficient backgrounds could be restored by hyperactivation of Rab2 GTPase mediated lysosome formation and fusion. These findings suggest that Hip14 and Patsas protect from the onset of HD like symptoms by acting as rate-limiting factors of lysosome formation and fusion.

## Introduction

Palmitoylation is a reversible post-translational lipid modification that alters the substrate proteins' hydrophobicity, thus affecting their intracellular trafficking, membrane association, stability and interactions with other proteins. Regulators of intracellular signaling and transport are over-represented among these substrate proteins [1,2], hence altered palmitoylation is implicated in the emergence of various diseases such as cancer, diabetes and neurological disorders [3]. The transfer of the palmitate chain to Cys residues of substrate proteins is catalyzed by DHHC (Asp-His-His-Cys) palmitoyl acyltransferase (PAT) enzymes, consisting of 4–6 transmembrane domains and a cytosolic Cys-rich domain (CRD) encompassing the eponymous DHHC catalytic motif. In addition, mammalian DHHC17/HIP14 (Huntingtin interacting protein 14) and DHHC13/HIP14L (Huntingtin interacting protein 14 like) harbor an additional N-terminal Ankyrin repeat domain (ANK), that interacts with and regulates the subcellular localization of several substrates involved in neuronal function, including the Huntington disease-associated protein Huntingtin (HTT) [4–6]. In Huntington's disease (HD), the CAG nucleotide expansion in HTT gene causes a poly-Q expansion in HTT protein which decreases its interaction with HIP14 and HIP14L, consequently, their enzymatic activity is lowered [5,7,8], presumably resulting in the mislocalization of multiple synaptic proteins [9]. In line with this, mice lacking HIP14 or HIP14L develop neuropathological phenotypes akin to HD [8,10,11]. The delivery of synaptic proteins is linked to the secretory pathway. Sorting of the secretory proteins with

different cellular destinations (plasma membrane, lysosome, secretory granules) takes place at the trans-Golgi network [12] and palmitoylation has been implicated as a key regulator of anterograde, plasma membrane-directed transport of proteins [13]. Since both HIP14 and HIP14L are Golgi resident enzymes, the mislocalization of synaptic proteins due to defective palmitoylation suggests that these PATs may have a critical role in maintaining proper protein sorting in the Golgi. However, how these PATs regulate post-Golgi membrane trafficking pathways is largely unexplored.

Drosophila is an excellent and popular in vivo experimental system for modeling human diseases (like HD or other neurodegenerative disorders) and understanding the underlying alterations in genetics and cell biology [14–16]. Moreover, the salivary gland of late L3 staged Drosophila larvae also became a powerful platform for studying the genetic regulation of post-Golgi trafficking, and particularly the complicated relationship of secretory and lysosomal pathways. Larval salivary gland cells synthesize and secrete high amounts of mucous Sgs (salivary gland secretion) or glue proteins in response to the molting hormone ecdysone [17–19]. The glue proteins at the trans-Golgi network are packaged into immature secretory granules [20,21], which undergo a complex maturation process involving a significant increase in size by homotypic fusions [22–25], progressive acidification [26–29], the profound remodeling of the secretory material [26–29] and the acquisition of membrane proteins required for exocytosis [24,30]. This is ensured by a series of fusion events between the maturing secretory granules and lysosomes that control the quantity and quality of the secretory material along the entire secretory pathway [31]. Hence, fusion of secretory granules with a heterogeneous non-degradative lysosomal population is critical for their maturation [27–29], while, the excess or abnormal secretory granules fuse with degradative lysosomes for selective degradation by crinophagy [28,29,32,33].

The Drosophila genome encodes two ANK domain-containing PATs: Hip14 (CG6017) and Patsas (CG6618), which are abundant in nervous tissues and localized to the Golgi apparatus [34]. Consistent with the mammalian results, Hip14 is essential for the proper development and efficient synaptogenesis and synaptic transmission by regulating the delivery of cysteine string protein (CSP) and synaptosome-associated protein 25 (Snap-25) by direct palmitoylation [35–37]. In contrast, Patsas is still poorly characterized. This raises the possibility that Hip14 (and potentially Patsas) has a conserved function in synaptic trafficking, but how these Golgi-localized ANK PATs control post-Golgi secretory pathways at cellular and molecular levels is still elusive. Here, we use the Drosophila larval salivary gland to understand the role of Hip14 and Patsas in the regulated secretory pathway and uncover their role in secretory granule-lysosome fusions and highlight their potential implications on neuronal function.

## Results

### 1. Patsas and Hip14 are important for proper crinophagic degradation of the content of secretory granules and their fusion with PI3P-positive endosomes

We chose the salivary gland of Drosophila larvae as a model system to study the role of ANK PATs in post-Golgi trafficking. Since maturation or crinophagic degradation of secretory granules requires the fine-tuned interplay of the secretory and lysosomal systems, these serve as good indicators of the integrity of post-Golgi trafficking. As Hip14 and Hip14L were identified as Golgi-resident enzymes in mammalian cells [38], we also verified the Golgi localization of Hip14 in larval salivary gland by co-staining Tomato-tagged Hip14 with Golgin-245 (S1 Fig). Since Golgi localization of Hip14 and Patsas were demonstrated in S2 cells [34] and for Hip14 also in the brain of adult flies by others [37,39], we conclude that, similarly to its human ortholog, Drosophila Hip14 and Patsas mostly function at the Golgi. We examined the potential role of the Patsas and Hip14 PAT enzymes in crinophagic degradation. At the first hour of puparium formation (~120 hrs after egg laying, hereafter referred as white prepupal stage/wpp), most glue granules are already secreted, and the non-secreted granules trapped in the cell fuse with degradative lysosomes to form crinosomes. To assess the efficiency of crinophagic degradation, we carried out a crinophagic flux assay by expressing the N-terminally GFP- and dsRed-tagged Sgs3 glue protein in secretory gland cells. The two reporters are equally sorted into the forming glue granules, so these appear as GFP and dsRed double-positive structures. After fusing with acidic lysosomes, the secretory granules are transformed into

degradative crinosomes in which the highly acidic milieu leads to the quenching of GFP (but not dsRed) signal, therefore the overlap between the reporters negatively correlates with the rate of crinophagic degradation, and mature crinosomes appear as dsRed-positive but GFP-negative structures [33]. We used RNAi-mediated knock-down of *Patsas* and *Hip14* to investigate their putative effect on the crinophagic flux. In contrast to the control cells which contain mostly dsRed-only crinosomes (58.7% of dsRed granules were negative for GFP) (Fig 1A and 1G-H), *Patsas* (Fig 1B) and *Hip14* (Fig 1C)

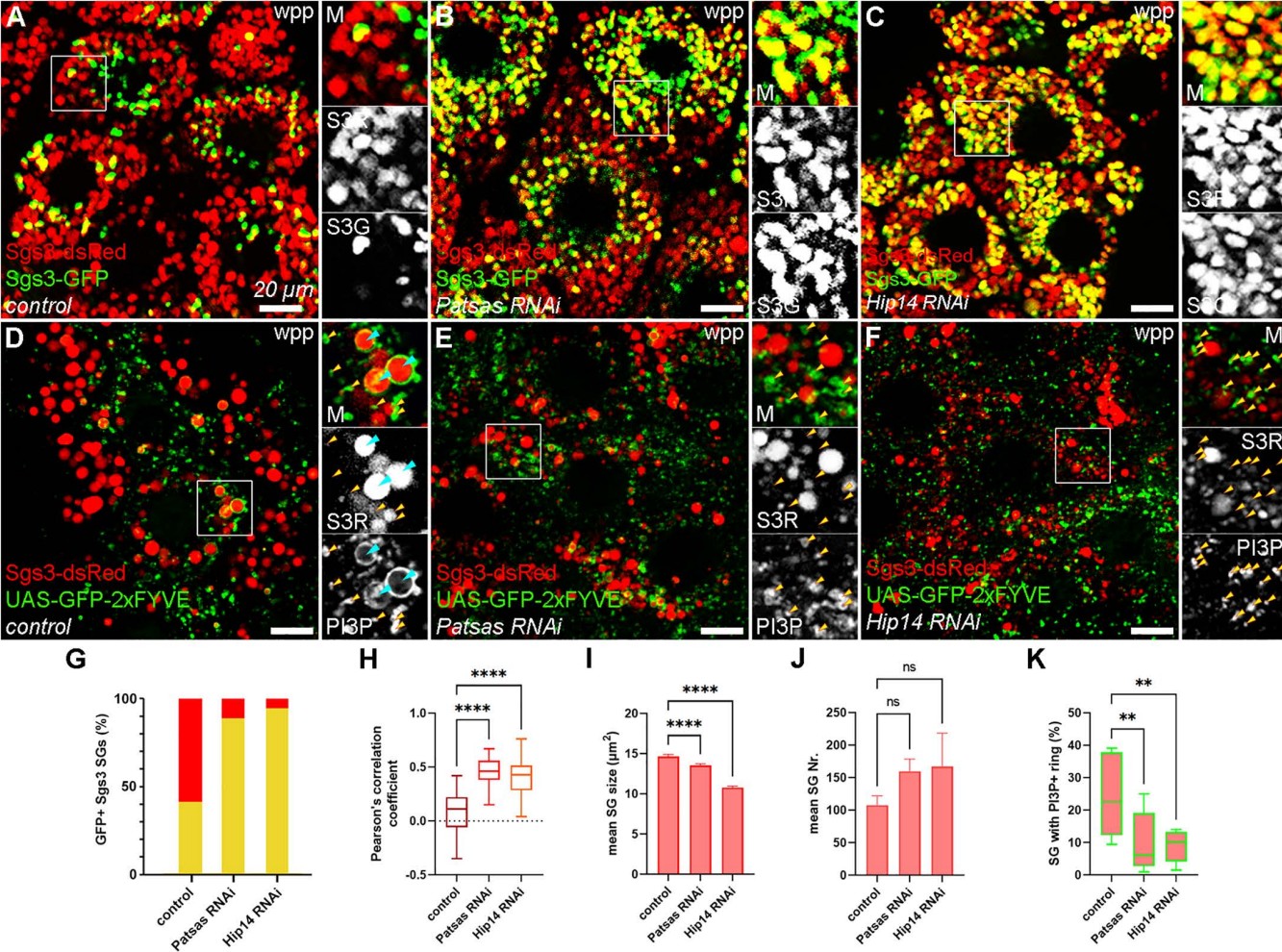

**Fig 1. Patsas and Hip14 are essential for proper crinophagic degradation. (A)** In control white prepupal (wpp) cells, the Sgs3-GFP signal is quenched in the acidic crinosomes. In contrast, the Sgs3-GFP signal is preserved in salivary glands of same age expressing Patsas **(B)** or Hip14 **(C)** RNAi. **(D)** In control cells, the formation of the crinosomes is supported by fusions between PI3P-positive endosomes (yellow arrowheads) and residual glue granules, resulting in the appearance of GFP-FYVE PI3P-specific reporter in a ring-like pattern around the crinosomes (turquoise arrowheads). In the absence of Patsas **(E)** or Hip14 **(F)**, these endosomal fusions are equally decreased and PI3P-positive endosomes accumulate among the glue granules. **(G-J)** Quantification of the data shown in **(A-C)** **(G)** Quantitative assessment of the proportion of randomly selected Sgs3-dsRed secretory granules with Sgs3-GFP positivity (n = 750 SGs from 5 cells of 3 different larvae. **(H)** Quantification of the overlap between the GFP- and dsRed-tagged Sgs3 reporters from n = 25 cells from 5 different pupae. **(I, J)** Quantification of Sgs3-dsRed SG size **(I)** and number **(J)** for the experiments shown in **(A-C)**. **(K)** Quantitative analysis of PI3P ring formation around Sgs3-dsRed positive secretory granules shown in **(D-F)**, n = 353 **(D)**, n = 428 **(E)**, n = 513 **(F)** glue secretory granules from 3 cells of 3 different larvae. Box plots indicate the range of data between the lower and upper quartiles, lines mark the median values, ****p < 0.0001, **p < 0.01, ns p > 0.05. Insets show 2x magnification of the outlined area, split into channels. Scale bar represents 20 μm in each panel. M: merged, S3G: Sgs3-GFP, S3R: Sgs3-dsRed, PI3P: phosphatidylinositol 3-phosphate (marked by GFP-2xFYVE probe), wpp: white prepupa, SG: secretory granule.

silenced cells accumulate double-positive intact glue granules, only 11.2% (Patsas RNAi) and 5.5% (Hip14 RNAi) of dsRed granules were GFP negative (Fig 1G-H). To confirm that this phenotype is not due to an off-target effect of the RNAi transgenes we repeated the same experiments with independent RNAi lines targeting Hip14 and Patsas, and they resulted in the same perturbation of crinophagic flux (S2A-C and S2G Fig). Since both Patsas and Hip14 RNAis caused a similar defect in crinophagy, we also tested whether these PATs are regulating distinct steps of the same pathway or acting redundantly with each other and carried out epistasis analyses. First, we knocked down both PATs simultaneously in the same salivary glands, which caused a similar perturbation in crinophagy as the single RNAis did (S2D and S2G Fig). Additionally, overexpression of Hip14 could rescue the defective flux phenotype of both the Patsas RNAis and Hip14 RNAis (S2E-G Fig). These results suggest that the two enzymes are rather regulating distinct steps of the same pathway and Hip14 may act downstream of Patsas.

In addition to their effect on crinophagic flux, we also observed that silencing of Patsas or Hip14 also resulted in a significant decrease in the size of Sgs3-dsRed positive structures, while their number has not changed significantly (Fig 1A-C and 1I-J). As these structures mostly represent crinosomes in control salivary glands at this developmental stage (white prepupae), this finding further suggests that Patsas and Hip14 deficient cells are defective for crinosome formation and the majority of smaller dsRed positive structures in them likely represent secretory granules (SGs) that cannot mature to crinosomes.

The glue granules at the white prepupal stage acquire phosphatidylinositol 3-phosphate (PI3P)-positive membranes presumably through fusion with endosomes, supporting the formation of crinosomes [28]. Along with the Sgs3-dsRed SG marker, we expressed the GFP-myc-2xFYVE PI3P-specific probe to investigate these endosomal fusions. In control cells, FYVE-positive rings appear around the Sgs3-dsRed-positive crinosomes (Fig 1D), indicating successful fusions with PI3P-positive endosomes. However, in the lack of Patsas (Fig 1E) and Hip14 (Fig 1F) PAT enzymes, these endosomal fusions are equally perturbed, and the PI3P+ endosomes accumulate among the glue granules, and the ring-like pattern of FYVE-GFP around the granules mostly disappears (Fig 1K). Thus, Patsas and Hip14 are similarly required for the fusion between the residual glue granules and PI3P-positive endosomes and consequently for the formation of crinosomes.

## 2. Patsas and Hip14 are required for the fusion of maturing secretory granules and lysosomes and their subsequent acidification

Since both Patsas and Hip14 proved to be required for the maturation of non-secreted SGs to crinosomes and their subsequent crinophagic degradation (at white prepupal stage), we aimed to test their potential effect on the maturation of the glue granules, preceding their bulk release. The fusion of the secretory granules with a heterogenous lysosomal compartment containing non-degradative lysosomes accompanies the complex maturation process of secretory granules, establishing their proper membrane composition and content for exocytotic release at the age of 2 hours before puparium formation (2h bpf) [27–29]. We analyzed the fusion between maturing glue granules and lysosomes in salivary gland cells expressing the N-terminally GFP-tagged Arl8 small GTPase, a well-established lysosome marker, and the Sgs3-dsRed reporters. In control cells, GFP-Arl8 positive rings appear around the Sgs3-dsRed structures, indicating successful SG-lysosome fusions (Fig 2A). In contrast, in the absence of Patsas (Fig 2B) or Hip14 (Fig 2C), Arl8-positive lysosomes accumulate among the glue granules, rather than forming rings around them (Fig 2A-C and 2E). Importantly, this fusion defect was not further exacerbated in Patsas, Hip14 double knockdown salivary glands (Fig 2D and 2E), indicating that these enzymes do not act redundantly in this process and both are required for efficient fusion between maturing secretory granules and lysosomes. In addition to the defective lysosomal fusion, the decreased size of maturing SGs was also detectable both in Patsas and Hip14 single and double deficient cells at 2h bpf, while SG numbers did not show any significant alterations in these genotypes (Fig 2F and 2G).

The gradual acidification accompanies the maturation of secretory granules, promoting the remodeling and processing of their secretory content to reach the appropriate consistency before exocytosis [26,27,29,40,41], and this acidification

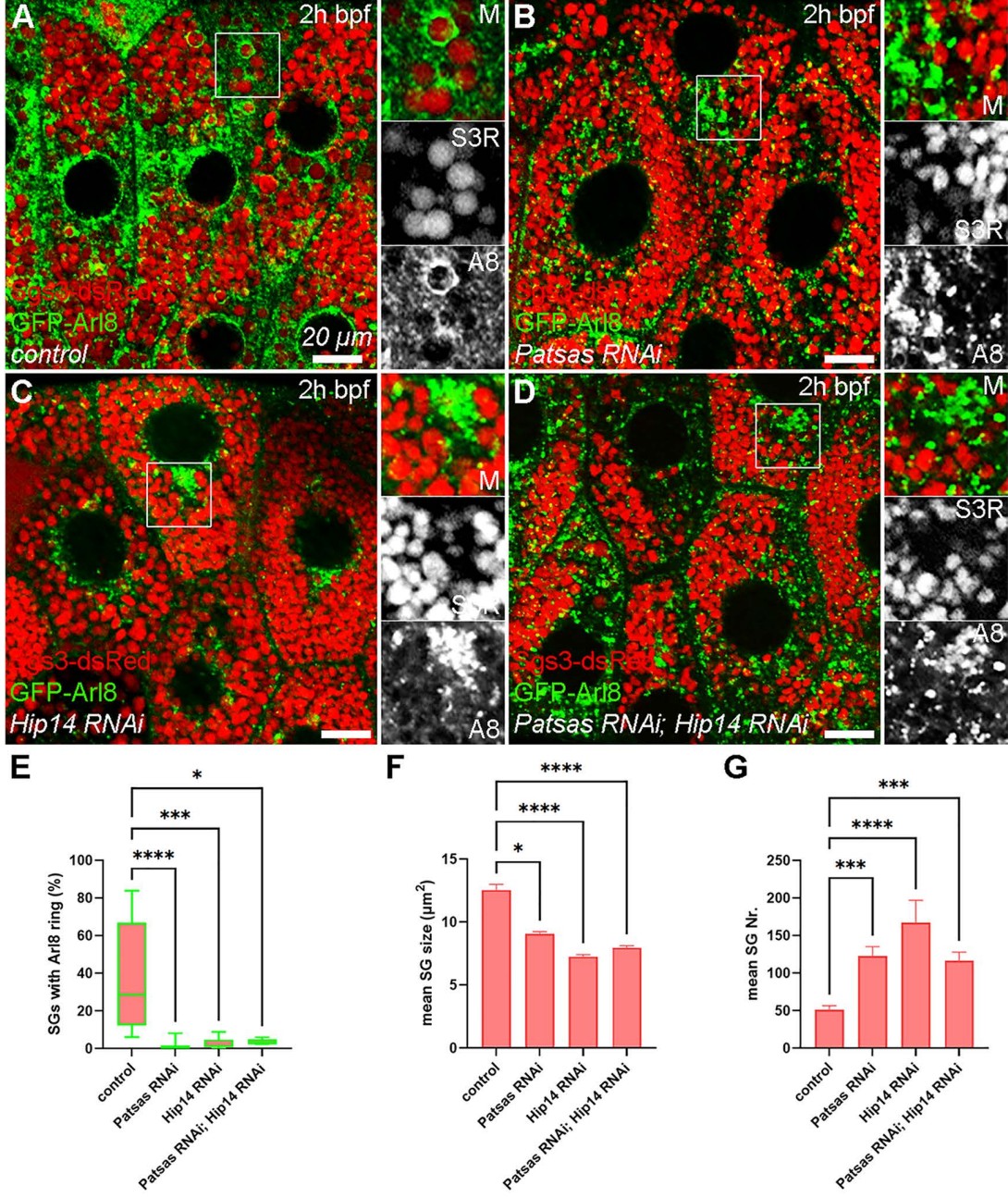

**Fig 2. Patsas and Hip14 are required for fusion between Arl8-positive lysosomes and maturing glue secretory granules. (A)** In control cells, Arl8-positive lysosomes fuse with maturing glue granules, resulting in the ring-like appearance of the reporter around their perimeter. In contrast, in the cells lacking either Patsas **(B)** or Hip14 **(C)**, or both **(D)**, Arl8-positive lysosomes accumulate among the maturing glue granules. **(E)** Quantitative analysis of the proportion of Sgs3-dsRed SGs bearing GFP-Arl8 reporter in **(A-D)**, n = 395 **(A)**, n = 1162 **(B)**, n = 971 **(C)**, n = 874 **(D)** glue granules from 3 cells of 3 different larvae, box plots represent the range of data between the lower and upper quartiles, lines mark the median. **(F, G)** Quantitative assessment of Sgs3-dsRed SG sizes **(F)** and numbers **(G)**, shown in **(A-D)**. ****$p < 0.0001$, ***$p < 0.001$, *$p < 0.05$, ns $p > 0.05$. Insets show 2x magnification of the outlined area, split into channels. Scale bar represents 20 µm in each panel. M: merged, A8: Arl8, S3R: Sgs3-dsRed, 2h bpf: 2 hours before puparium formation, SG: secretory granule.

PLOS Genetics

is highly dependent on secretory granule-lysosome fusions [28]. To assess whether the acidification of maturing glue granules is affected by ANK PATs, we stained salivary gland cells expressing Sgs3-dsRed with LysoTracker deep Red (LTdR) vital dye that selectively labels acidic structures like lysosomes and crinosomes. In control salivary glands, large LTdR-positive structures appear at the age 2h bpf, and these are also positive for Sgs3-dsRed, indicating efficient SG-lysosome fusion at this stage (Fig 3A). In contrast, Patsas (Fig 3B) and Hip14 (Fig 3C) deficient cells, the mean size but not the number of LTdR positive structures, significantly drops (Fig 3A-C, 3E, and 3F), and small punctate lysosomes appear, suggesting that loss of these ANK PATs critically reduces the efficiency of SG-lysosome fusions. To evaluate whether overactivation of Golgi palmitoylation may lead to an opposite effect, we overexpressed Hip14 in same aged salivary glands and we observed a remarkable increase in LTdR signal intensity (Fig 3D and 3G), however, the size and number of these structures did not alter compared to the controls (Fig 3A and 3D-F). Hence, we conclude that Patsas and Hip14 are required for proper acidification of maturing secretory granules, and the extent of acidification positively correlates with the level of Hip14. Interestingly, the average size of the SGs significantly dropped, but their number remained unaltered, in Hip14 overexpressing cells (Fig 3D, 3H, and 3I), suggesting that Hip14 mostly supports SG-lysosome heterotypic fusions, and does not facilitate SG growth by promoting SG-SG homotypic fusions.

Acidification, chloride and calcium ion uptake drive the structural remodeling of the content of maturing glue granules and crinosomes, which can be monitored at the ultrastructural level [26,27,29]. In control cells at the 2h bpf stage, large mature granules can be observed with multiple electron-dense cores (Fig 4A). Although in the lack of Patsas (Fig 4B) or Hip14 (Fig 4C) the glue granules show similar morphology, fusion-incompetent or docked lysosomes frequently appear in their vicinity. In contrast, salivary gland cells overexpressing Hip14 show small immature secretory granules with the premature presence of crinosomes with an unusual multivesicular or multilamellar morphology and containing glue-like material in a highly disintegrated form (Fig 4D). Importantly, these premature crinosomes likely represent the acidic structures labeled very intensively with LTdR (Fig 3D). These results further support that Patsas or Hip14 PAT enzymes are required for and act as rate-limiting factors of heterotypic fusions between maturing secretory granules and lysosomes.

### 3. ANK PAT mediated heterotypic fusions negatively regulate Sgs3 secretion

As both overexpression and silencing of ANK-PATs resulted in reduced SG size, we further analyzed the fate of SGs by testing the effect of Patsas and Hip14 on glue secretion. We applied a simple but well-established experimental setup: In control white prepupae secreted glue can be detected as an Sgs3-dsRed positive trace on the surface of pupae that are adhered to the wall of the vial, while the non-secreted glue appears as residual dsRed signal in the salivary glands (Fig 5A and 5E). Interestingly, although the signal of secreted glue was also detectable in Patsas (Fig 5B and 5E) and Hip14 RNAi (Fig 5C and 5E) white prepupae, the dsRed signal was weaker in the salivary glands. In contrast, Hip14 overexpression (Fig 5D and 5E) caused reduced glue secretion accompanied by extensive retention of Sgs3-dsRed in salivary glands. These findings suggest that SG release is negatively regulated by ANK-PATs and the smaller SG size detectable in both ANK-PAT deficient and Hip14 overexpressing cells can be likely attributed to increased secretion or accelerated crinophagic elimination of SGs, respectively.

### 4. Patsas and Hip14 are involved in the biosynthetic transport of lysosomal hydrolases

As the presence of both ANK PATs was critical for proper fusion of lysosomes with secretory granules, we were interested in whether these Golgi-resident enzymes are also required for lysosomal biosynthetic transport and function. To assess whether these PATs are general regulators of lysosome acidification, we carried out additional LysoTracker Red (LTR) stainings on salivary glands that express Lamp1-GFP late endosome/lysosome reporter in which the GFP moiety is located on the luminal side of lysosome membrane and exposed to the acidic milieu [42]. Hereby, in control cells, the large LysoTracker-positive structures of the size of glue granules appear, in which the Lamp1-GFP signal was mostly quenched, and the two markers barely colocalized with each other (S3A Fig). However, in the lack of Patsas (S3B Fig) or Hip14

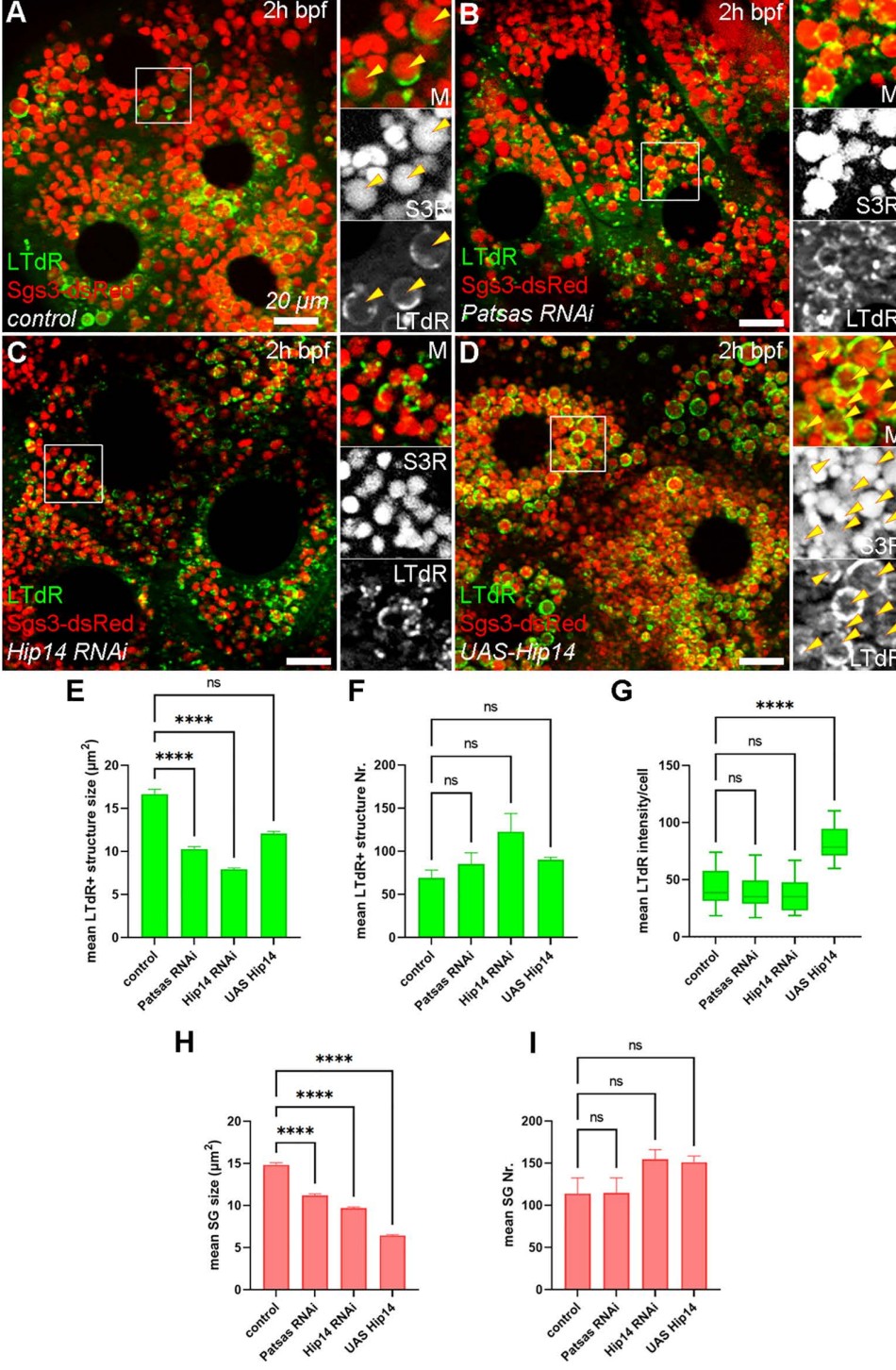

**Fig 3. Patsas and Hip14 are required for proper maturation and acidification of secretory granules. (A)** In control salivary gland cells, Sgs3-dsRed-positive mature secretory granules become acidic, and positive for LysoTracker deepRed dye (LTdR) (yellow arrowheads). However, in the absence of Patsas **(B)** or Hip14 **(C)**, smaller LTdR+ structures and small lysosomes appear. **(D)** In contrast, in Hip14 overexpressing cells, the smaller Sgs3-positive structures are intensively labeled with LTdR, indicating enhanced acidification. **(E-I)** Quantification of the data shown in **(A-D)**. **(E-F)** Quantitative evaluation of the size **(E)** and number **(F)** of LTdR+ structures in **(A-D)**, n = 1038 **(A)**, n = 1282 **(B)**, n = 1843 **(C)**, n = 1353 **(D)** LTdR+ structures from 5 cells of 3 different larvae, error bars mark ± SEM. **(G)** Quantification of mean LTdR intensities, n = 15 cells from 3 different salivary glands. Box

plots indicate the range of data between the lower and upper quartiles, lines mark the median. **(H-I)** Quantitative analysis of the size **(H)** and number **(I)** of Sgs3-dsRed-positive structures, n = 2846 **(A)**, n = 2870 **(B)**, n = 3869 **(C)**, n = 3781 **(D)** Sgs3 + structures from 5 cells of 5 different larvae, error bars mark ± SEM. ****p < 0.0001, ns p > 0.05. Insets show 2x magnification of the outlined area, split into channels. Scale bar represents 20 μm in each panel. M: merged, S3R: Sgs3-dsRed, 2h bpf: 2 hours before puparium formation, SG: secretory granule.

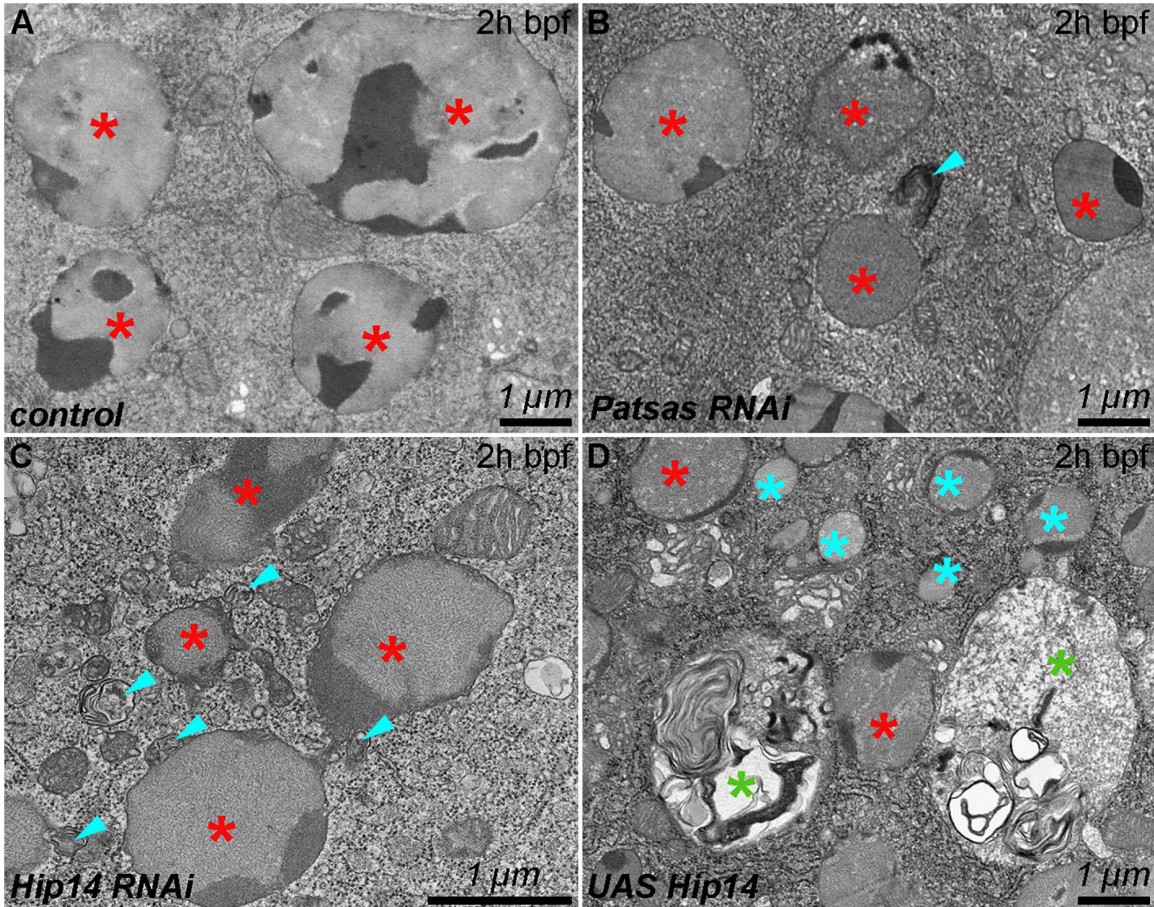

**Fig 4. Patsas and Hip14 are required for the effective fusion of lysosomes with maturing glue granules. (A)** In control cells, large maturing glue granules (red star) with multiple electron-dense cores can be observed shortly before bulk glue secretion (2h bpf). **(B-C)** In same-aged salivary glands that are deficient for Patsas **(B)** or Hip14 **(C)**, mature glue granules appear with multilamellar (fusion incompetent) lysosomes (turquoise arrowheads) at their edge. **(D)** In cells overexpressing Hip14, numerous small-sized immature glue granules (turquoise star) appear, together with a premature appearance of abnormal multivesicular crinosomes (green star). Scale bar represents 1 μm in each panel. 2h bpf: 2 hours before puparium formation.

(S3C Fig), the LTR-positive structures are smaller in size and frequently overlap with Lamp1-GFP, representing fusion-defective and less acidic lysosomes, while the number of LTR structures did alter in these S3E and S3F Fig. In contrast, overexpression of Hip14 (S3D Fig) caused a massive accumulation of large LysoTracker positive structures that were labelled with the dye in a higher intensity compared to control cells (S3D-G Fig) and were devoid of Lamp1-GFP.

To assay the putative role of Patsas and Hip14 enzymes in the sorting and transport of lysosomal proteins, we immunostained the lysosomal protease Cathepsin L in cells expressing the Lamp1-GFP reporter. In control cells, Cathepsin L showed moderate colocalization with Lamp1-GFP, especially on smaller vesicles (Fig 6A). In contrast, the lack of Patsas

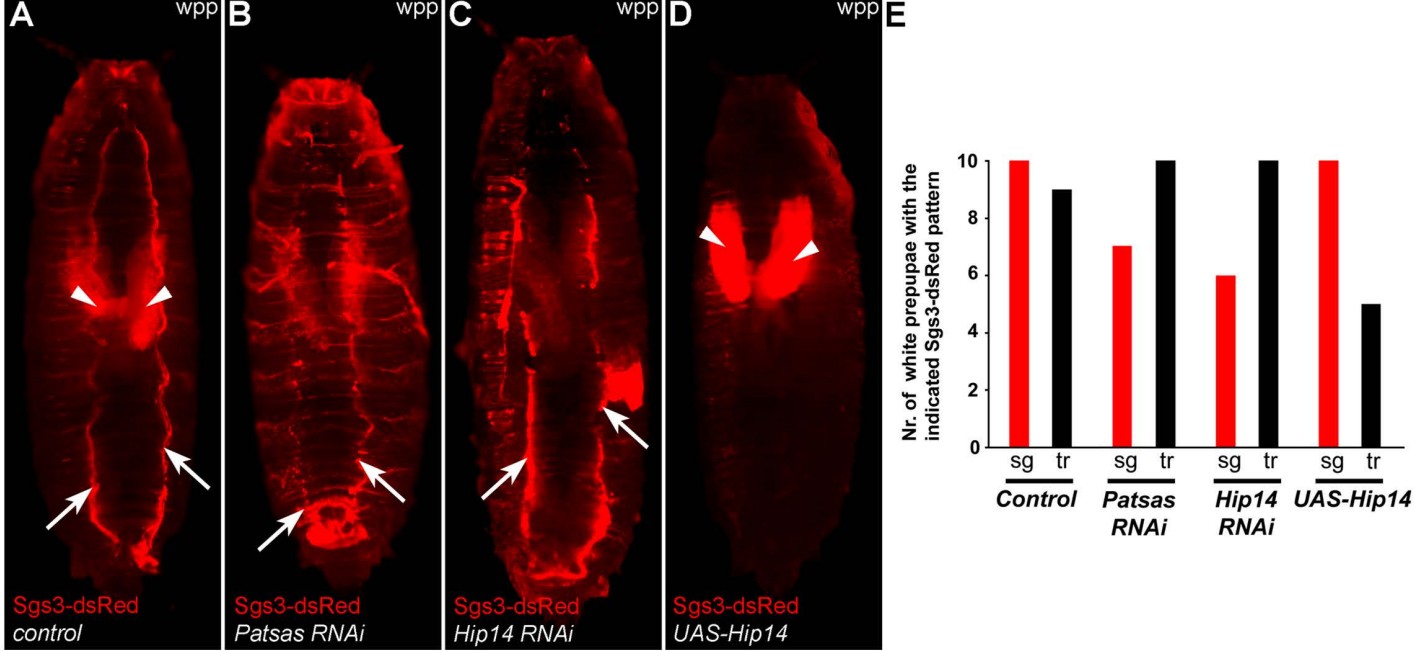

**Fig 5. Patsas and Hip14 negatively affect the exocytotic discharge of glue secretory granules. (A)** In control white prepupae (wpp), most of the glue secretory material is released and expelled from the salivary glands and appears as traces on the ventral side of the pupae (white arrows), while some residual granules are retained in salivary gland (white arrowheads) cells. In the absence of Patsas **(B)** or Hip14 **(C)**, the glue signal in salivary glands is decreased. **(D)** In contrast, the overexpression of Hip14 led to an increase in the salivary gland retained glue signal, which is also accompanied by a decline in the ventral glue traces. **(E)** Quantification of the data shown in **(A-D)**, n = 10 prepupae were assessed per genotype. sg: salivary gland, tr: glue traces, wpp: white prepupa.

(Fig 6B) or Hip14 (Fig 6C) equally led to the decrease in the size of Cathepsin L-positive structures, that became more puncta-like (Fig 6A-C and 6E) and resulted in a more extensive colocalization between the two markers. Importantly, this size reduction was not accompanied by increased abundance of Cathepsin L positive structures (Fig 6A-C and 6F), suggesting the impairment of the formation and integrity of lysosomal compartment in ANK PAT deficient tissues. Finally, the overexpression of Hip14 increased the size, but not the number of Cathepsin L structures (Fig 6D-F). Since these enlarged Cathepsin L positive structures did not colocalize with Lamp1-GFP, it suggests that they likely represent the very acidic, multivesicular crinosomes (Figs 4D and S3D) where the Lamp1-GFP reporter gets degraded rapidly.

To follow the transport of lysosomal enzymes at the ultrastructural level, we carried out Gömöri's acid phosphatase (AcPase) enzyme cytochemistry, which enables us to follow the subcellular localization of lysosomal AcPase enzyme activity. Glue granules gradually acquire the AcPase enzyme during their maturation [27,29], possibly by fusion with non-degradative lysosomes. We carried out this assay on salivary glands of larvae at different developmental stages. We observed that immature glue granules of control cells at the early L3 larval stage (6h bpf) contain a few acid-phosphatase-positive precipitates (S4A Fig). Later, shortly before bulk glue secretion (2h bpf), more AcPase signal is present in mature glue granules (S4B Fig). In the lack of Patsas or Hip14 PAT enzymes, the glue granules and the crinosomes contain less AcPase precipitate throughout their maturation (S4C-F Fig), accompanied by the simultaneous accumulation of AcPase-containing multilamellar/multivesicular lysosomes at the rim or proximity of the granules at 2h bpf stage (S4D and S4F Fig). In contrast, in cells overexpressing Hip14, AcPase-positive crinosomes appear already at the earliest (6h bpf) stage and have an irregular multivesicular morphology (S4G Fig). Interestingly, crinosomes that form prematurely fall into two groups: those that mostly contain glue only show more AcPase precipitate, while those with multivesicular morphology

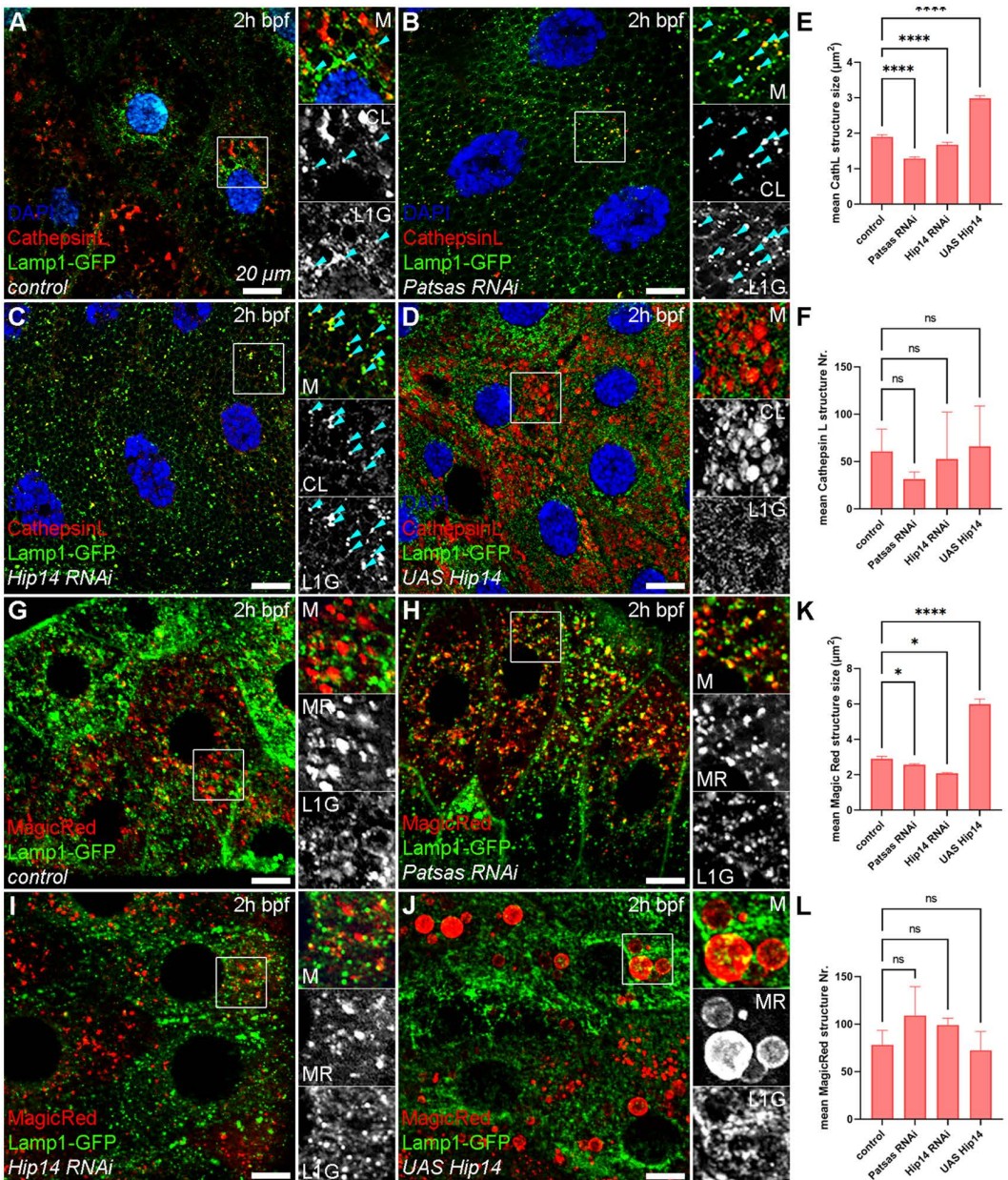

**Fig 6. Patsas and Hip14 are essential for proper transport of Cathepsin L lysosomal hydrolase. (A)** In control cells, the lysosomal protease Cathepsin L colocalizes with Lamp1-GFP on small lysosomes (turquoise arrowheads), but not on larger Cathepsin L positive structures. The absence of Patsas **(B)** or Hip14 **(C)** equally led to the puncta-like appearance of the Cathepsin L-positive lysosomes, which show more extensive overlap with Lamp1-GFP. **(D)** In contrast, the overexpression of Hip14 significantly increases the size of Cathepsin L structures, which do not overlap with Lamp1-GFP. **(E-F)** Quantification of the size **(E)** and number **(F)** of Cathepsin L-positive structures shown in **(A-D)**, n = 1523 **(A)**, n = 788 **(B)**, n = 1318 **(C)**, n = 1650 **(D)** Cathepsin L+ structures from 5 cells of 5 different larvae. **(G)** In control cells, maturing secretory granules and small lysosomes exhibit Cathepsin L activity, as indicated by MagicRed fluorescence. In Patsas **(H)** or Hip14 **(I)** knocked-down cells, these MagicRed-positive structures are smaller. **(J)** In contrast, the overexpression of Hip14 led to the enlargement of MagicRed-positive structures. **(K, L)** Quantification of the size **(K)** and number **(L)** of MagicRed-positive structures showed in **(G-J)**, n = 1176 **(G)**, n = 1638 **(H)**, n = 1487 **(I)**, n = 1089 **(J)** MagicRed+ structures from 5 cells of 3 different larvae, error bars mark ±SEM. ****p < 0.0001, *p < 0.05, ns p > 0.05. Insets show 2x magnification of the outlined area, split into channels. Scale bar represents 20 µm in each panel. M: merged, CL: Cathepsin L, L1G: Lamp1-GFP, MR: MagicRed, 2h bpf: 2 hours before puparium formation.

have less but still detectable amount of AcPase precipitate and reduced glue content, presumably due to their enhanced degradation (S4G Fig). At the 2h bpf stage, mostly immature glue granules with moderate AcPase signal can be detected and the multivesicular crinosomes have only residual AcPase content (S4H Fig). These results demonstrate that the lysosomal enzyme transport is severely impaired in the lack of Patsas or Hip14 palmitoyl-transferases, which consequently delays the maturation of glue granules too. In contrast, the overexpression of Hip14 was able to enhance the fusion of lysosomes with glue granules, resulting in the premature appearance of Cathepsin L and AcPase-positive crinosomes. Taken together, our data suggests that although lysosomal hydrolases can reach the lysosomal compartment even in the absence of Patsas or Hip14, the proper transport and subcellular distribution of lysosomal enzymes is critically influenced by ANK PATs.

Finally, to evaluate whether ANK PAT dependent alterations in lysosomal trafficking also affects enzymatic activity of lysosomal hydrolases, we stained salivary glands at the 2h bpf age with MagicRed, an endocytic tracer that becomes fluorescent upon Cathepsin L-mediated cleavage in the lumen of actively degrading lysosomes. Control tissues (Fig 6G) show two populations of MagicRed positive structures: first the smaller puncta-like fraction that were also positive for Lamp1-GFP likely represent immature lysosomes or late endosomes, while the larger, Lamp1-GFP negative ones represented mature, actively degrading lysosomes and possibly some mature SGs. However, this latter fraction is missing from Patsas (Fig 6H) and Hip14 (Fig 6I) deficient tissues, which contain significantly smaller MagicRed positive structures, without any alteration of their number (Fig 6K and 6L). In contrast, Hip14 overexpression (Fig 6J-L) gives rise to enlarged MagicRed positive structures that are devoid of Lamp1-GFP, which likely represent large mature lysosomes and premature crinosomes. This data clearly points out that the level of ANK-PATs proportionally regulates lysosomal function, including biosynthetic transport and SG-lysosome fusion.

To further characterize the molecular mechanism how ANK PATs regulate lysosome morphology and function, we selected three proteins, huntingtin (Htt) [5,6,43], PI4KIIα [2,44] and Vamp7 [45] that are involved in post-Golgi and endo-lysosomal membrane trafficking and are putative targets of HIP14/Hip14- or HIP14L/Patsas- mediated palmitoylation. To assay whether the loss of these proteins similarly perturbs lysosomal morphology as ANK PAT RNAis, we immunolabelled Cathepsin L in salivary glands at 2h bpf age. Compared to the controls (S5A Fig) no alteration of Cathepsin L structure size or number was observed in Htt (S5B, S5E, and S5F Fig) or PI4KIIα (S5C, S5E, and S5F Fig) deficient cells, while Vamp7 (S5D- S5F Fig) knock-down cells showed smaller, puncta-like lysosome morphology. Hence, Vamp7 may be considered as a good candidate to serve as potential target protein through which ANK-PATs regulate lysosomes' function and fusion capability.

## 5. Expression of the constitutively active form of Rab2 can compensate for the lysosomal defects seen in Patsas or Hip14 deficient cells

The Golgi-localized Rab2 small GTPase protein promotes the biogenesis of the Golgi-derived biosynthetic vesicles and mediates their fusion with lysosomes of different (endocytotic, secretory or autophagic) origin [46–48]. Moreover, overexpressing the constitutively active form of Rab2 was shown to increase glue granule size and their fusion with lysosomes [33]. We sought to examine whether the reduced volume and dysfunctionality of the lysosomal compartment caused by the lack of these PAT enzymes could be restored by ectopic stimulation of lysosomal biogenesis and fusions. Therefore, we expressed a constitutively active Rab2[Q65L] point mutant allele in Sgs3-dsRed expressing salivary gland cells and performed LTdR staining. In control cells, glue granules are properly acidified and are positive for LTdR (Fig 7A and 7E) while, as expected, the expression of the Rab2[Q65L] promoted the formation of the glue granules and their fusion with acidic lysosomes, thus increasing the size of the LTdR-positive maturing glue granules (Fig 7B and 7E), accompanied by an accumulation of smaller (presumably immature) glue granules (Fig 7B and 7F-G). Importantly, hyperactivation of Rab2 causes a similarly increased size of these LTdR-positive glue granules even in the absence of Patsas (Fig 7C and 7E) or Hip14 (Fig 7D and 7E) compared to the control. As expression of Rab2[Q65L] also decreased the mean size of SGs (Fig

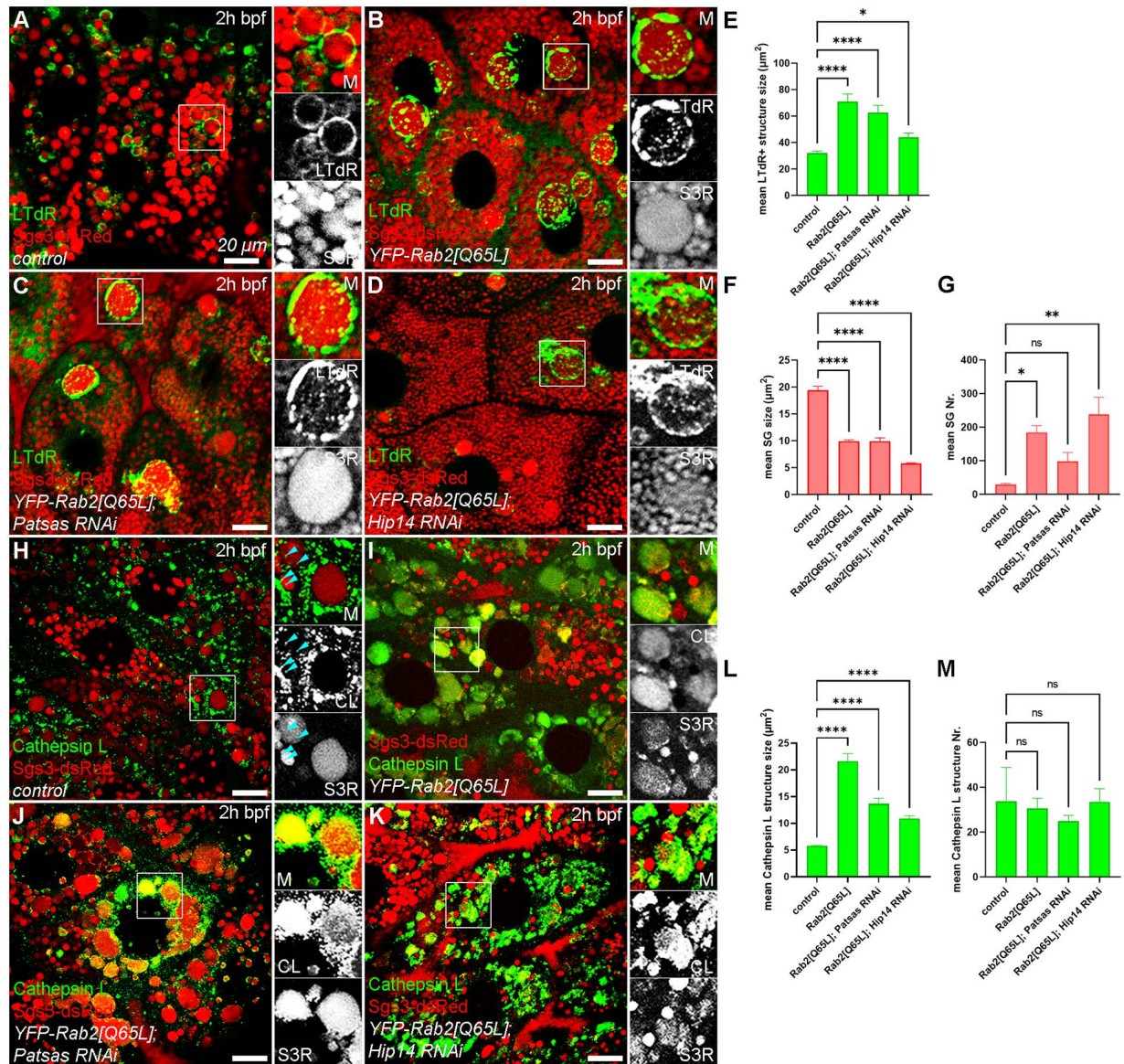

**Fig 7. Hyperactivation of Rab2 could compensate for defective acidification and Cathepsin L transport caused by the absence of Patsas and Hip14. (A)** In control cells, maturing glue granules progressively acidify and become positive for LysoTracker deepRed (LTdR). **(B)** The expression of the constitutively active Rab2[Q65L] transgene in control background increases the size of the LTdR-positive secretory granules. **(C, D)** The expression of Rab2[Q65L] can similarly increase the size of LTdR-positive glue granules even in the absence of Patsas **(C)** or Hip14 **(D)** PAT enzymes compared to the control **(A)**. **(E-G)** Quantification of the data shown in **(A-D)**, 5 cells of 3 different larvae were analyzed per genotype. **(E)** Quantitative assessment of LTdR-positive structure sizes, n = 150 randomly selected LTdR-positive structures. **(F, G)** Quantitative analysis of the size **(F)** and number **(G)** of Sgs3-dsRed secretory granules, n = 448 **(A)**, n = 2767 **(B)**, n = 1475 **(C)**, n = 3570 **(D)** SGs. **(H)** In control cells, Cathepsin L-positive lysosomes appear among the maturing glue granules, which also contain some small Cathepsin L-positive foci (turquoise arrowheads). **(I)** The expression of Rab2[Q65L] in control cells enhances the fusion of Cathepsin L-positive lysosomes with maturing glue granules and increases their size. **(J, K)** Rab2[Q65L] overexpression can also increase the size of Cathepsin L-positive structures in Patsas **(J)** and Hip14 **(K)** deficient genetic background. **(L, M)** Quantification of the size **(L)** and number **(M)** of Cathepsin L-positive structures shown in **(H-K)**, n = 843 **(H)**, n = 765 **(I)**, n = 623 **(J)**, n = 838 **(K)** Cathepsin L-positive structures from 5 cells of 5 different larvae. Error bars mark ± SEM, ****p < 0.0001, **p < 0.01, *p < 0.05. Insets show 2x magnification of the outlined area, split into channels. Scale bar represents 20 μm in each panel. M: merged, CL: Cathepsin L, LTdR: LysoTracker deepRed, S3R: Sgs3-dsRed, 2h bpf: 2 hours before puparium formation, SG: secretory granule.

7F) and inhibited Glue secretion (S6A and S6C Fig) its effect is reminiscent of that we observed upon Hip14 overexpression. Moreover, constitutively active Rab2 could also decrease the hypersecretory phenotype of Hip14 deficient background (S6B and S6C Fig). However, as Rab2[Q65L] also increased the SG number in any genotypes (although it was not significant in Patsas RNAi), likely through promoting their formation (Fig 7G), it seems, that hyperactivation of Rab2 or Hip14 may not act through the same mechanism. Thus, our findings suggest promoting Golgi exit and lysosomal fusions by Rab2 hyperactivation can mediate a compensatory effect on perturbed glue granule acidification caused by ANK PAT deficiency.

Next, we investigated whether the smaller size of Cathepsin L positive structures in Patsas and Hip14 deficient cells could be similarly reversed by activation of Rab2. Therefore, we carried out immunolabelings with Cathepsin L-specific antibodies on control and Rab2[Q65L] overexpressing salivary glands. In control cells, Cathepsin L-positive lysosomes are mostly positioned among the glue granules, and a few small Cathepsin L positive foci could be also observed inside the granules (Fig 7H) that indicates moderate level of glue granule-lysosome fusion. The overexpression of constitutive active (Rab2[Q65L]) (Fig 7I and 7L) but not the wild type (Rab2[WT]) form of Rab2 (S7A, S7B, and S7H Fig) increases the size of the Cathepsin L-containing glue granules by enhancing the fusion of Cathepsin-L positive lysosomes and maturing glue granules, which results in the accumulation of Cathepsin L in these granules (Fig 7I and 7L). Importantly, these Cathepsin L-positive granules also increased in size even in the absence of the Patsas (Fig 7J and 7L) and Hip14 (Fig 7K and 7L) enzymes compared to the control cells (Fig 7H and 7L) upon the overexpression of Rab2[Q65L] but not a Rab2[WT] (S7A-D and S7H Fig). Importantly, we did not detect any significant changes in the number of Cathepsin L positive structures upon Rab[Q65L] overexpression (Fig 7M). These data suggest that the perturbed lysosomal function and impaired glue granule-lysosome fusion in ANK PAT deficient cells (Fig 6B and 6C) could be not only restored but also overcompensated by ectopically enhancing Rab2 mediated lysosomal biogenesis and fusion.

As loss of Vamp7 decreased the size of Cathepsin L, similarly to the loss of ANK PATs (S5D and S5E Fig), we also investigated whether overexpression of Vamp7 may have an opposite effect on lysosome morphology. Similarly to the effect of the Rab2[Q65L] (Fig 7I and 7L), overexpression of a GFP-Vamp7 transgene also increased the size of Cathepsin L positive structures and promoted their overlap with Sgs3-dsRed positive SGs both in control (S7A, S7E, and S7H Fig) and Patsas (S7F and S7H Fig) or Hip14 deficient (S7G and S7H Fig) backgrounds.

## 6. Hip14 is required for proper lysosome formation in the adult brain and for adult locomotor performance

As the lack of mammalian Hip14 causes HD-like neurological disorders, we assessed the effect of the absence of Hip14 and Patsas on the integrity of the lysosomal system in neurons and the neuromuscular function of adult flies. We temporally controlled the expression of *Hip14* and *Patsas* RNAi transgenes by combining neuron-specific Appl-Gal4 and a temperature-sensitive tub-Gal80 allele [49] to avoid early undesirable lethality [36,37] and induced the expression of the RNAi transgenes only in adult flies. First, we immunostained 21-day-old adult brains with Cathepsin L antibody to analyze the integrity of the lysosomal compartment. Interestingly in brains lacking Patsas or Hip14, a moderate but significant increase in the size, but not the number of Cathepsin L-positive structures can be detected (Fig 8A-E), indicating that loss of ANK PATs dysregulates lysosome formation in these neurons. Additionally, we also carried out LysoTracker Red stainings on 21-day-old control and Hip14 or Patsas deficient brains that also expressed Lamp1-GFP. Importantly, as Lamp1-GFP is generally quenched in acidic, actively degrading lysosomes we saw only a moderate overlap between these two markers in control brains (Fig 8F). In contrast, and in line with our observations in 2h bpf salivary glands (S3B and S3C Fig), LTR and Lamp1-GFP extensively colocalized in Patsas (Fig 8G) and Hip14 (Fig 8H) RNAi samples and, similarly to Cathepsin data (Fig 8A-E), the increased size without any changes in number of Lamp1- (Fig 8I and 8J) and LysoTracker- (Fig 8K and 8L) positive structures was also detectable in these genotypes, indicating that loss of these enzymes leads to serious disregulation of lysosome morphology and function.

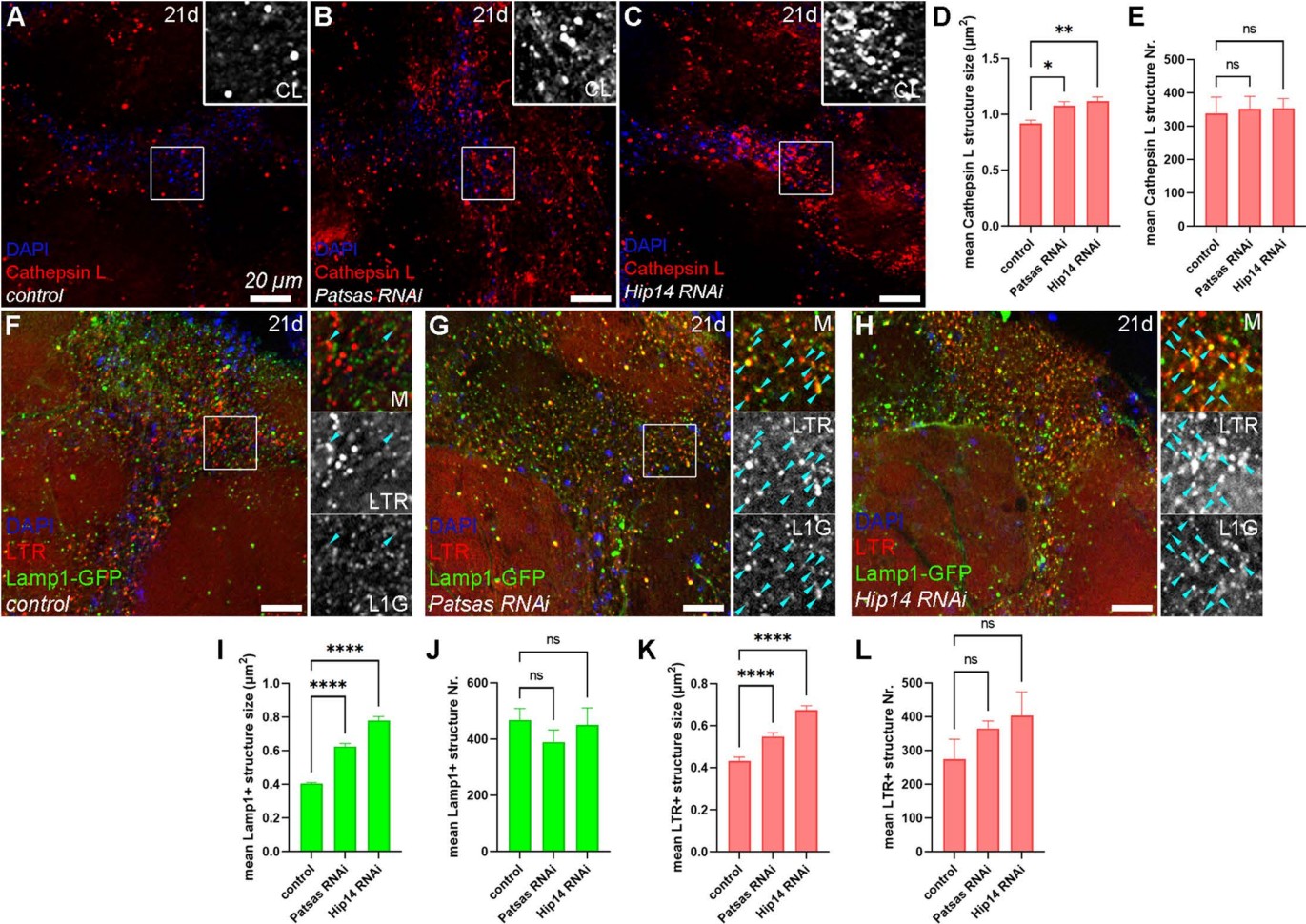

**Fig 8. Patsas and Hip14 are essential for proper lysosomal function in the adult brains. (A-C)** Compared to control brains **(A)**, neuron-specific loss of either Patsas **(B)** or Hip14 **(C)** leads to a moderate increase in the size of Cathepsin L-positive lysosomes. **(D, E)** Quantification of the size **(D)** and number **(E)** of Cathepsin L-positive lysosomes in 21 days-old (21d) adult (female) brains, n = 1692 **(A)**, n = 1762 **(B)**, n = 1772 **(C)** Cathepsin L-positive lysosomes from 5 different brains. **(F)** In control brains, the Lamp1-GFP signal is quenched in the acidic, LysoTracker Red-positive lysosomes, only a few double-positive lysosomes (turquoise arrowheads) can be observed. However, in the absence of Patsas **(G)** or Hip14 **(H)** the LTR-positive lysosomes increase in size and retain the Lamp1-GFP fluorescence. **(I, J)** Quantitative assessment of the size **(I)** and number **(J)** of Lamp1+positive structures, n = 2344 **(F)**, n = 1951 **(G)**, n = 2256 **(H)**. **(K, L)** Quantitative analysis of the size **(K)** and number **(L)** of LTR+ structures, n = 1373 **(F)**, n = 1829 **(G)**, n = 2017 **(H)**. Error bars mark ± SEM, **** p < 0.0001, **p < 0.01, *p < 0.05, ns p > 0.05. Insets show 2x magnification of the outlined area, split into channels. Scale bar represents 20 µm in each panel. M: merged, CL: Cathepsin L, L1G: Lamp1-GFP, LTR: LysoTracker Red.

We performed a climbing test to examine the effect of neuron-specific loss of Hip14 or Patsas on adult locomotion. In contrast to the controls, the Hip14 knock-down flies show a progressive decline in climbing ability (that can be detected from day 14). Importantly, the observed ataxia in Hip14 could be restored by the expression of the constitutively active Rab2 transgene (Figs 9 and S8) in both sexes. Interestingly, Patsas deficient flies also showed a progressive defect in climbing phenotype (Figs 9 and S8). These results indicate that Hip14 and Patsas are required for the maintenance of neuronal function, likely through regulating the morphology and activity of lysosomes. Furthermore, our results suggest that the ectopic activation of lysosomal biogenesis and fusions could be a good strategy to compensate for the lack of Hip14 and to restore proper neuronal function.

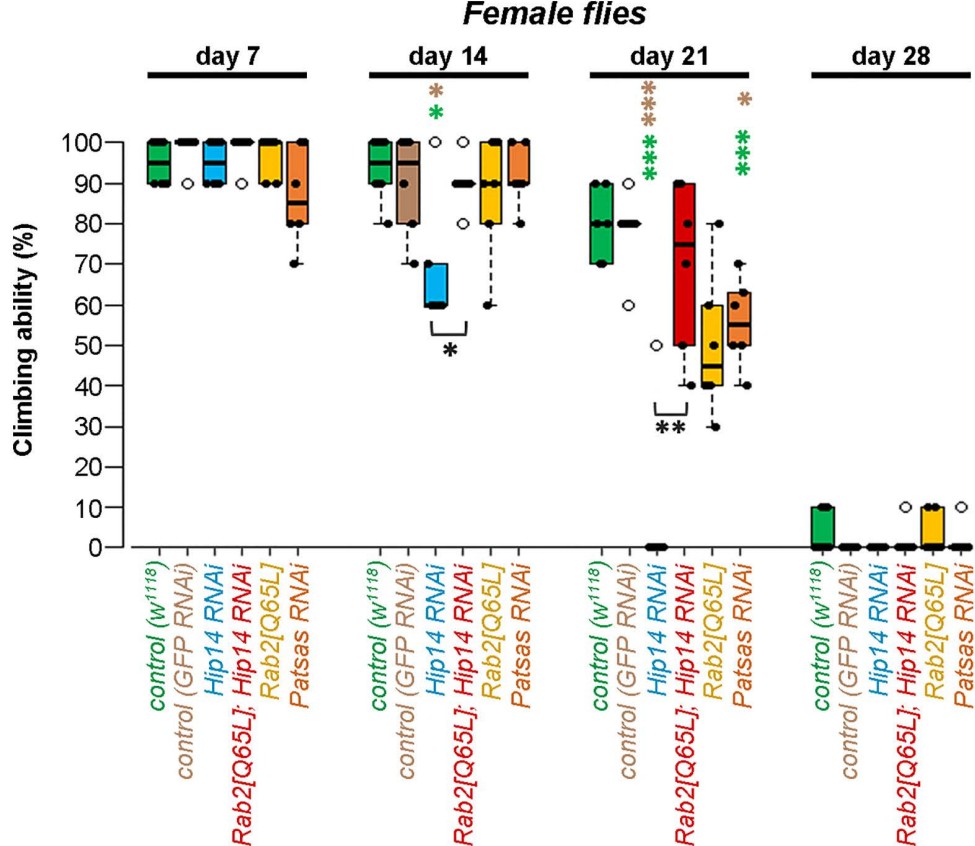

**Fig 9. Patsas and Hip14 are required for proper neuromuscular performance in female adult flies.** The brain-specific knock-down of Hip14 caused a progressive decline in the motor performance of adult flies, compared to the controls ($w^{1118}$ and *GFP RNAi*) in 14 days-old female flies. The constitutively active form of Rab2 was able to restore the climbing defect of flies lacking Hip14. The brain-specific silencing of Patsas also caused a significant decline in neuromotor performance in 21-days-old flies, compared to the controls. Black dots represent one measurement, circles indicate outliers, boxes represent the typical 50% of the climbed adults, the lines show the median and whiskers present the upper and lower quartiles, n = 20 flies per genotype. Significant differences are indicated by ***$p < 0.001$, **$p < 0.01$, *$p < 0.05$, significant difference compared to control ($w^{1118}$) and control (GFP RNAi) flies respectively are represented by green and brown *-s. Otherwise significant differences between two genotypes are represented by black *-s over the clasps.

## Discussion

In our work, we found that both Patsas and Hip14 are critical regulators of the interplay between lysosomes and the secretory pathway. First, we revealed that these enzymes contribute to efficient crinophagic degradation and fusions between residual glue granules and PI3P-positive endosomes, aiding the formation of crinosomes [28]. Second, we demonstrated that both ANK PATs promote the fusion of maturing glue granules with non-degradative lysosomes, which facilitates their progressive acidification and ultrastructural remodeling [26–29]. Third we found that loss of Hip14 also perturbs lysosomal morphology and function both in the larval salivary gland and the adult brain and leads to gradual impairment in neuromotor performance. Finally, these lysosome-related phenotypes caused by loss of ANK PATs could be partially restored by ectopic activation of lysosomal fusion.

The hypothesis that Patsas and Hip14 can promote lysosomal fusions, and regulate lysosomal protein sorting is further supported by a recent study where Hip14 has emerged as a regulator of *Drosophila* host defense against bacterial infection, presumably by maintaining lysosomal homeostasis [50]. It is important to note that this recent study did not explore

the functionality and fusion capabilities of the affected lysosomes. In our work, we made significant efforts to find out how modulation of ANK PATs affects lysosomal maturation and fusion by assaying the localization of two different lysosomal enzymes. As a result, we demonstrated that loss of either Patsas or Hip14 leads to a significant decrease in fusion between maturing glue granules and Arl8-positive lysosomes and the LysoTracker-positive acidic compartment, resulting in reduction in size of LysoTracker positive structures, likely representing fusion incompatible lysosomes with a decreased acidification capacity. Moreover, we also observed the decline in size of Cathepsin L-positive lysosomes, decreased lysosomal hydrolytic activity by MagicRed and impaired delivery of lysosomal AcPase to maturing glue granules in ANK PAT deficient secretory cells. Vice versa, the overexpression of Hip14 resulted in the expansion of LysoTracker and Cathepsin L-positive structures, which likely represent premature crinosomes that were also positive for AcPase. These findings suggest that the activity of Golgi resident ANK PATs may serve as a rate-limiting factor in lysosome maturation, function and their fusion with SGs (Fig 10). This function of Patsas and Hip14 is further confirmed by our finding that enhancing the fusion of lysosomes with Golgi-derived vesicles or secretory granules in an alternative way [33,46] – by overactivation of Rab2 GTPase – could compensate for lysosomal dysfunction in Patsas or Hip14 deficient cells.

How ANK PATs can regulate lysosomal fusions at the molecular level is still elusive. Several studies showing that SNARE proteins [51,52], and especially the lysosomal R-SNARE Vamp7 [45] – a known critical player in secretory granule-lysosome fusion [28,33] – can be also palmitoylated. By testing the loss of function phenotypes of Vamp7 with other putative targets of HIP14/HIP14L homologs Htt and PI4KIIα, we found that Vamp7 could be a good candidate among these to be considered as a potential palmitoylation-target of Patsas or Hip14 through which these enzymes affect lysosome function and fusions. This is further suggested by our epistasis analysis where Vamp7 overexpression could reverse the effect of ANK-PATs on lysosome size. However, how palmitoylation alters Vamp7's function in a molecular level and whether Vamp7 can be palmitoylated by Patsas and Hip14 is beyond the scope of this current work and yet to be elucidated by further studies. Additionally, we cannot entirely rule out that the elevated level of lysosomal content in secretory granules in Hip14 overexpressing cells is partially due to the missorting of lysosomal enzymes into the secretory granules, and not only a result of elevated lysosomal fusions. Understanding whether Hip14 has a direct effect on the sorting of lysosomal enzymes in the Golgi requires further research.

We successfully demonstrated that loss of Patsas and Hip14 also perturbs the lysosomal system in the brain, which is accompanied by declined locomotor performance, a characteristic phenotype of HD-like neurodegeneration [53]. Although, lysosome function and acidification similarly changed in ANK PAT deficient salivary glands and brains, the lysosome morphology altered in opposite directions in these tissues. This might be due to tissue specific differences in the role of the lysosomal compartment, as lysosomes are mostly required for supporting the secretory processes in salivary glands, while this may be less prominent in brain. Most importantly, the poor neuromuscular performance caused by Hip14 deficiency could be restored by expressing the constitutively active form of Rab2, consistent with our previous results. A growing amount of evidence suggests that lysosomal dysfunction is a likely cause for a large cohort of neurodegenerative disorders [54,55], and it was also shown recently that ectopic activation of lysosome fusion in some disease models can ameliorate their symptoms [56]. HD is accompanied by dysfunctional autophagy and defective autophagosome-lysosome fusion [57–59]. Our current research supports that HD-like symptoms can be caused, at least in part, by perturbation of lysosome fusions. Moreover, since HTT was shown to be required for the proper activity of HIP14 and HIP14L [5,7,8], our findings that Hip14 and Patsas acts as rate-limiting factors in lysosome fusions may provide a straightforward mechanistic explanation of how mutant HTT leads to lysosomal dysfunction.

One interesting finding of our work is that the loss of Patsas or Hip14 alone is enough to cause a similar perturbation in post-Golgi trafficking and lysosomal fusions. Our epistasis analysis where Patsas and Hip14 double deficient cells did not showed further perturbation in crinophagic flux than single knock downs did, suggests that these enzymes act non-redundantly. On the other hand, as ectopic expression of Hip14 could rescue both the lack of Patsas or Hip14 it rather suggests that these enzymes may act in the same pathway and likely on the same substrates, but Hip14 is functionally

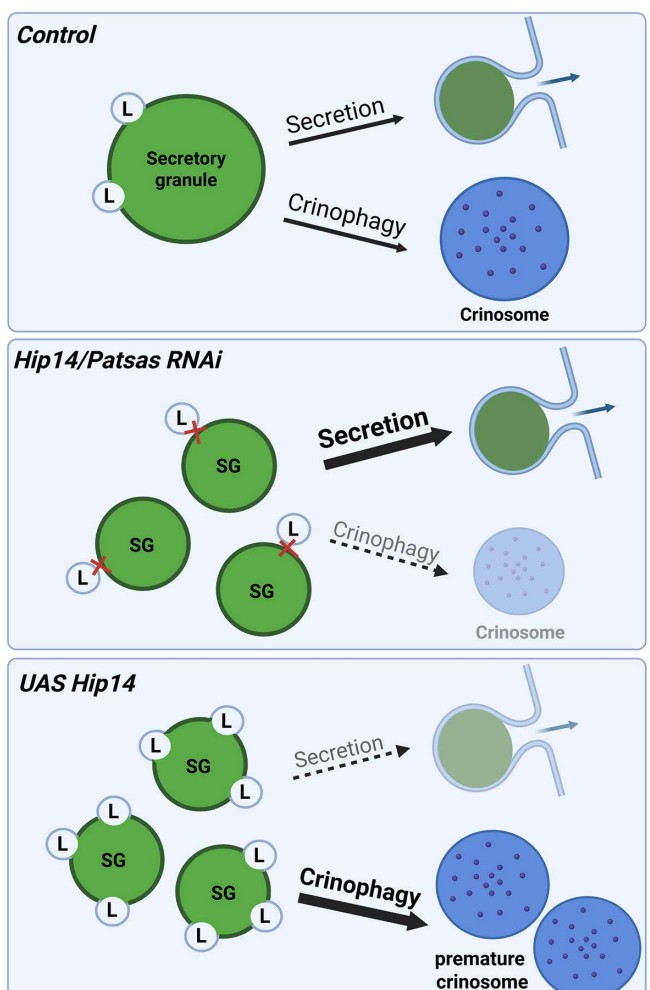

**Fig 10. Model of the function of Hip14 and Patsas in secretory granule-lysosome fusion and maturation of glue secretory granules.** In control cells, moderated amount of fusion events between lysosomes and glue secretory granules (SGs) contributes to proper maturation and release of SGs by exocytosis, while the residual SGs form crinosomes by fusion with degradative lysosomes, where the secretory material can be degraded in a regulated fashion. In the absence of Hip14 or Patsas, both lysosomal function and secretory granule-lysosome fusion are hampered, thus the formation of crinosomes is inhibited and the majority of the glue secretory granules are directed towards exocytotic discharge. In contrast, the overexpression of Hip14 can enhance the fusion of lysosomes and maturing secretory granules, leading to premature and increased formation of crinosomes, while preventing the exocytotic release of the secretory granules.

downstream of Patsas. An explanation for this could be that the activity of Hip14 and Patsas is closely related, as has been observed between Hip14 and other DHHC enzymes [13,60,61]. For example, one of the enzymes binds substrates stronger, but the other performs the palmitoylation more efficiently, as suggested for mammalian Golgi PATs [13,62,63]. However, we also cannot rule out an alternative, although less likely explanation that Golgi has such a high demand of PAT activity [13], so the cell cannot tolerate the lack of either enzymes. Further research should be conducted to uncover what the actual relationship is between the two ANK PATs' activities and substrate preference.

In conclusion, our data show that the two Ankyrin-repeat DHHC palmitoyl-transferase enzymes, Patsas and Hip14 are essential for the proper maturation and crinophagic degradation of the secretory granules by mediating their fusion with the endo-lysosomal compartment. Furthermore, the activity of both PAT enzymes is equally required for proper lysosomal

function by regulating lysosomal fusion events and mediating the transport of various lysosomal hydrolases. Similarly, the presence of Patsas and Hip14 is also essential for the proper lysosomal function and neuronal health in the central nervous system, and the defects caused by the absence of Hip14 both in secretory cells and neurons could be partially restored by tissue-specific overexpression of constitutively active Rab2. Our results demonstrate that restoration of lysosome maturation and fusions may be a good strategy to overcome the lack of the HD-associated PAT, Hip14 and to restore neuronal function.

## Materials and methods

### Drosophila genetics

The flies were raised at 25°C temperature on a standard yeast-cornmeal-agar medium. The w[1118] (#3605), fkh-Gal4 (#78060), appl-Gal4 (#32040), tubP-Gal80[ts] (#7017), UAS-YFP-Rab2[Q65L] (#9760), UAS-GFP-myc-2xFYVE (#42712), UAS-Hip14[WT] (#42697), Sgs3-GFP (#5884), UAS-Patsas[JF01773] RNAi (#31218), UAS-PI4KIIα[HMC06016] RNAi (#65110), UAS-VAMP7[GL01524] (#43543) RNAi, UAS-GFP[VALIUM22] RNAi (#41551) and UAS-Hip14[JF01167] RNAi (#31591) (referred as Hip14 RNAi/2 in the text) stocks were obtained from Bloomington Drosophila Stock Center (Bloomington, US). The UAS-Hip14[6017R-1] RNAi (referred as Hip14 RNAi in the text) and UAS-Patsas[6618R-1] RNAi (referred as Patsas RNAi/2 in the text) lines were purchased from NIG-Fly (National Institute of Genetics, Japan). UAS-htt[KK100789] RNAi was obtained from Vienna Drosophila Resource Center (Vienna, Austria). The Sgs3-dsRed reporter was kindly provided by Andrew Andres (University of Nevada, US), the UAS-Lamp1-GFP reporter by Helmut Krämer (University of Texas Southwestern Medical Center, US), UAS-GFP-Arl8 line by Sean Munro (MRC Laboratory of Molecular Biology, UK), UAS-Hip14-tdTomato by Steve Stowers (Montana State University, US) and UAS-GFP-Vamp7 by Amy Kiger (University of California, US). To avoid early lethality caused by the absence of *Hip14*, a temperature-sensitive tubP-Gal80 construct was used to temporally regulate the expression of transgenes in the climbing assay. These flies were initially kept at 25°C temperature during metamorphosis and then the hatched adults were placed at the 29°C restrictive temperature.

### LysoTracker staining

The larval salivary glands were dissected in cold PBS (pH = 7.4) and gently permeabilized for 30 s in 0.05% Triton X-100-PBS (PBTX) solution. The glands were rinsed in PBS (3x30 s) and incubated for 2 min in 0.5 μM LysoTracker Red or 1 μM LysoTracker deep-Red (dissolved in PBS, Invitrogen) solution, then washed in PBS (3x30 s) and mounted with 9:1 PBS:glycerol media containing 1 μg/ml DAPI (4′,6-diamidino-2-phenylindole, Sigma Aldrich) as nuclear stain. The 21-daysold adult brain samples were dissected in ice-cold PBS and incubated for 15 min in 0.5 μM LysoTracker Red-PBS solution, rinsed in PBS (3x30 s) and mounted with the same DAPI-containing 9:1 PBS:glycerol media.

### MagicRed staining

The larval salivary gland cells were dissected in ice-cold PBS (pH = 7.4), gently permeabilized in 0.05% PBTX solution (30s) and rinsed in PBS (3x30s). The glands were incubated in Cathepsin L substrate MagicRed staining solution (1:26, BioRad #6137, 10 min RT). The samples were washed and mounted with 9:1 PBS:glycerol media containing 1 μg/ml DAPI.

### Immunocytochemistry

The larval salivary glands were dissected in cold PBS, then permeabilized with 0.05% PBTX solution for 30 s and fixed in 4% formaldehyde solution (dissolved in PBS, 40 min, RT). Then, the glands were washed with PBS (3 × 5 min, RT), incubated in a blocking solution (5% fetal calf serum in 0.1% PBTX, 30 min, RT) and incubated with the first antibodies (dissolved in blocking solution, overnight, 4°C). Then the samples were washed (PBTX, 3x10 min) and incubated in

blocking solution (30 min, RT) and with the secondary antibodies (diluted in blocking solution, 3h, RT, protected from light). After that samples were incubated in 4% NaCl solution (15 min, RT), which contained Hoechst (1:200, Sigma-Aldrich) nuclear dye and washed (2 × 15 min in 0.1% PBTX, then 3 × 15 min in PBS). Finally, the samples were dissected and mounted in 100% glycerol. The adult brains were fixed in 4% formaldehyde solution (30 min), washed with PBS (3x10 min, RT), blocked (10% fetal calf serum in 0.1% PBTX, 3 hours, RT) and incubated with the first antibodies (in blocking solution) for 2 consecutive nights (4°C), then with the secondary antibodies (ON, 4°C). The samples were treated with the Hoechst-containing 4% NaCl solution, washed (same as for salivary glands) and mounted in clear glycerol placed between spacers with the antennal side facing upside.

For the immunostainings, chicken anti-GFP (1:1000, Thermo Fisher) and rabbit anti-CathL/MEP (abcam, #ab58991, 1:100) primary and the AlexaFluor488-conjugated anti-chicken and AlexaFluor634- and AlexaFluor568-conjugated anti-rabbit secondary antibodies (Invitrogen, 1:000) were used.

## Fluorescent imaging

Fluorescent images were taken at room temperature with an AxioImager M2 microscope equipped with an ApoTome.2 structured illumination unit (Zeiss, Germany), Orca-Flash 4.0 LT3 digital sCMOS camera (Hamamatsu Photonics, Japan), EC Plan-Neofluar 2.5x/0.075, EC Plan-Neofluar 20x/0.50, Plan-Apochromat 40x/0.95 and Plan-Apochromat 63x/1.4 Oil objectives (Zeiss, Germany). Raw images were processed with ZEN2.3 lite Microscopy Software (Zeiss) and Photoshop CS4 (Adobe Systems, US).

## Electron microscopy

The larval salivary glands were fixed in 1% glutaraldehyde, 2% formaldehyde, 3mM $CaCl_2$, 1% D-saccharose in 0.1N sodium cacodylate buffer (pH = 7.4, ON, 4°C). The samples were washed with 0.1N sodium-cacodylate buffer and incubated in 0.5% $OsO_4$ (1h, RT) and half-saturated uranyl acetate (30 min, RT), gradually dehydrated in ethanol with increasing concentration and embedded in Durcupan (Fluka) according to the manufacturer's protocol. The ultrathin (ca. 70nm) sections were contrasted in Reynold's lead citrate. The samples for Gömöri's acid phosphatase histochemistry were dissected in 2% glutaraldehyde, 2% formaldehyde containing EM fixative (ON, 4°C), and the final lead contrasting was omitted, otherwise we followed the protocol described before [64].

## Climbing tests

We measured the climbing ability of the 7-, 14-, 21- and 28-days old fruit flies (kept at 29°C). Two parallel tubes were used for each measurement. We anesthetized each genotype with CO2, then placed 10–10 females in each tube. The animals then had a rest period of 1.5 hours at 29°C to avoid the aftereffect of CO2. Following a period of rest, the climbing test was repeated three times, with a 30-minute interval between each trial. Each experimental trial was recorded by a camera and subsequently analyzed to assess the motor performance of each genotype. The long-term climbing experiment is conducted to the final boundary line (22 cm) on the glass tube, with the objective being to ascertain the number of animals that reach this boundary line within 40 seconds of each genotype [65].

## Statistics

For the quantitative analysis of fluorescent images, the ImageJ software (National Institutes of Health, Bethesda, Maryland, US) was used. The overlap between markers was determined by Pearson's correlation analysis using the Coloc2 plugin. The size of the LysoTracker and Cathepsin L structures were analyzed in the 0.5-99 $\mu m^2$ size range to get rid of the background noise and the irrelevant large, coalesced structures with a size of multiple glue granules. The threshold for these images was set by the same person. The enlarged Cathepsin L-positive structures were counted in the relevant

3–500 µm² range, while the LysoTracker deep-Red structures were manually selected. Fluorescent intensities per single cells for the LysoTracker experiments were measured in Image J software using the freehand ROIs, excluding the nuclei to restrict the measures to the cytosol and lysosomal compartment. For each ROI, the mean gray value was extracted as a measure of average fluorescence intensity per cell. Background correction was performed for all measurements by subtracting the intensity of a background ROI of equal area and the background-corrected values were processed using Python. For the quantitative assessment of ring-like structure formation, the glue granules were manually selected and the percentage of them with ring-like structures was analyzed. The small lysosomes in adult brains were analyzed within the same 100x100 µm area between the antennal lobe and the mushroom body lobe with 0.1-10 µm² size range. For comparison of datasets following Gaussian distribution, one-way ANOVA with Tukey's post-hoc test (Figs 1H, 1J, 1K, 3F, 3G, 6L, 7G, 7M, 8E, 8J, 8L, S2G, S3F and S3G) and if at least one of the datasets followed a non-Gaussian distribution, Kruskal-Wallis test (Figs 1I, 2E, 2F, 2G, 3E, 3H, 3I, 6E, 6F, 6K, 7E, 7F, 7L, 8D, 8I, 8K, S3E, S5E, S5F, and S7H) was performed. For pairwise comparison of datasets following Gaussian distribution, two-tailed t-test and if one dataset followed non-Gaussian distribution, Mann-Whitney U-test was performed (Figs 9 and S8). The statistical analyses were performed with the GraphPadPrism 9.0.0 software (Boston, Massachusetts, US) and for climbing tests with RStudio (v2023.06.2). Elements of Fig 10 were created in BioRender. Juhasz, G. (2025) s2v31gt https://BioRender.com/. All quantification raw data that was used for statistics can be found in S1 Table.

## Supporting information

**S1 Fig. *Drosophila* Hip14 localizes to the Golgi apparatus.** The Hip14-Tomato reporter extensively overlaps (turquoise arrowheads) with the trans-Golgi-specific Golgin-245 marker in control larval salivary gland cells. Insets show 2x magnification of the outlined area, split into channels. Scale bar represents 20 µm. M: merged, G245: Golgin-245, 2h bpf: 2 hours before puparium formation.
(TIF)

**S2 Fig. Patsas and Hip14 are required for crinophagic degradation, and their absence could be rescued by Hip14 expression. (A)** In control salivary gland cells of white prepupae (wpp), most of the Sgs3-GFP signal is quenched in the acidic milieu of the crinosomes; only a few intact double-positive granules can be observed. In contrast, the Sgs3-GFP signal is equally preserved in salivary gland cells expressing independent RNAi transgenes for Patsas **(B)** and Hip14 **(C)** or both **(D)**. **(E, F)** The overexpression of Hip14 could rescue the compromised crinophagic degradation in the absence of either Patsas **(E)** or Hip14 **(F)**. **(G)** Quantification of the overlap between the GFP- and dsRed-tagged Sgs3 reporters from n = 25 cells of 5 different larvae, box plots indicate the range of data between the lower and upper quartiles, lines mark the median, ****p < 0.0001, ns p > 0.05. Scale bar represents 20 µm in each panel. wpp: white prepupa.
(TIF)

**S3 Fig. Patsas and Hip14 are required for proper lysosomal acidification. (A)** In control salivary gland cells, large LysoTracker Red-positive (LTR+) acidic structures of the size of glue granules appear, in which the GFP signal of the Lamp1-GFP reporter is quenched. **(B-C)** In the absence of Patsas **(B)** or Hip14 **(C)**, smaller LTR+ structures appear with retained GFP fluorescence (turquoise arrowheads). **(D)** In contrast, the overexpression of Hip14 causes the enlargement of LTR+ acidic structures. **(E, F)** Quantitative assessment of the size **(E)** and mean number **(F)** of the LTR+ structures shown in **(A-D)**, n = 2182 **(A)**, n = 2199 **(B)**, n = 3853 **(C)**, n = 2921 **(D)** LTR+ structure from 5 cells of 5 different larvae, error bars mark ± SEM. **(G)** Quantification of mean LTR intensities in **(A-D)**, box plots indicate the range of data between the lower and upper quartiles, lines mark the median. ****p < 0.0001, ***p < 0.001, ns p > 0.05. Insets show 2x magnification of the outlined area, split into channels. Scale bar represents 20 µm in each panel. M: merged, L1G: Lamp1-GFP, LTR: LysoTracker Red, 2h bpf: 2 hours before puparium formation.
(TIF)

**S4 Fig. Patsas and Hip14 are required for proper transport of acidic phosphatase enzyme into maturing glue secretory granules. (A)** The small immature granules (turquoise star) in control cells of wandering (6h bpf) larvae contain very few acidic phosphatase (AcPase) precipitates (appears as black dots). **(B)** The mature glue granules (red star) shortly prior to the bulk glue secretion (2h bpf) contain more AcPase. **(C)** In the absence of Patsas, immature glue granules contain similar level of AcPase signal to the age-matched control, while highly AcPase-reactive lysosomes (turquoise arrowheads) can be observed in the proximity of the granules. **(D)** 2h bpf stage mature glue granules have a moderate AcPase signal compared to the control, while AcPase-containing (fusion incompetent) lysosomes accumulate among them. **(E)** The immature glue granules of cells lacking Hip14 show similar AcPase enzyme activity to the control, while the mature glue granules at 2h bpf stage contain slightly less which is accompanied by the appearance of highly AcPase-positive lysosomes at their proximity **(F)**, similar to the *Patsas* RNAi **(D)**. **(G)** In contrast, the immature glue granules of Hip14 overexpressing cells contain a lot of AcPase precipitate, with the premature appearance of crinosomes. These crinosomes split into two groups, those with the conventional content containing much (yellow star) and those with an irregular multivesicular morphology (green star) with less AcPase positivity. **(H)** Later at the 2h bpf stage Hip14 overexpressing cells contain AcPase-positive immature glue granules. Scale bar represents 1 μm in each panel. 6h bpf: 6 hours before puparium formation, 2h bpf: 2 hours before puparium formation.
(TIF)

**S5 Fig. Vamp7, unlike Htt and PI4KIIα, is involved in proper transport of Cathepsin L lysosomal hydrolase. (A)** In control salivary gland cells, Cathepsin L partially colocalized with the Lamp1-positive lysosomes. The absence of Htt **(B)** or PI4KIIα **(C)** does not impact the size of the Cathepsin L-positive structures. However, the lack of Vamp7 **(D)** significantly decreases the size of Cathepsin L-positive structures, similarly to Patsas and Hip14. **(E, F)** Quantification of the size **(E)** and number **(F)** of Cathepsin L-positive structures from 5 cells of 5 different larvae, n = 2992 **(A)**, n = 3722 **(B)**, n = 2275 **(C)**, n = 3549 **(D)** Cathepsin L-positive structures. Error bars mark ± SEM, ****p < 0.0001, ns p > 0.05. Insets show 2x magnification of the outlined area, split into channels. Scale bar represents 20 μm in each panel. M: merged, CL: Cathepsin L, L1G: Lamp1-GFP, 2h bpf: 2 hours before puparium formation.
(TIF)

**S6 Fig. The overexpression of the constitutively active form of Rab2 inhibits the exocytotic release of glue granules. (A)** The exocytotic release of Sgs3-dsRed-positive glue granules is compromised in the salivary gland cells of white preupae expressing the constitutively active form of Rab2 small GTPase, thus the glue secretory material is retained in salivary glands (white arrowheads) and absent from traces. **(B)** The exocytotic release of glue granules is similarly inhibited in salivary gland cells that simultaneously express the constitutively active form of Rab2 and the Hip14 RNAi transgenes. **(C)** Quantification of the data shown in **(A-B)**, n = 10 different white prepupae per genotype. wpp: white prepupa.
(TIF)

**S7 Fig. The overexpression of Vamp7, but not the wild-type Rab2, can increase the size of the Cathepsin L-positive structures even in the absence of Patsas or Hip14. (A-D)** In control cells **(A)**, most Cathepsin L-positive lysosomes are located among the secretory granules and overexpression of wild-type Rab2 in a control background **(B)** does not alter the size of Cathepsin L-positive structures, which remain smaller in the absence of Patsas **(C)** or Hip14 RNAi **(D)**. **(E-G)** In contrast, the overexpression of Vamp7 **(E)** promotes fusion between Cathepsin L-positive lysosomes and maturing granules, thereby increasing their size, even in the absence of Patsas **(F)** and Hip14 **(G)**. **(H)** Quantification of the size of Cathepsin L-positive structures shown in **(A-G)**, n = 6165 **(A)**, n = 4260 **(B)**, n = 4277 **(C)**, n = 3076 **(D)** n = 1110 **(E)**, n = 613 **(F)**, n = 759 **(G)** Cathepsin L-positive structures from 5 cells of 5 different larvae. Error bars mark ± SEM, ****p < 0.0001, ns p > 0.05. Insets show 2x magnification of the outlined area, split into channels. Scale bar represents 20μm in each panel. M: merged, CL: Cathepsin L, S3R: Sgs3-dsRed, 2h bpf: 2 hours before puparium formation.
(TIF)

**S8 Fig. The absence of Patsas and Hip14 are required for proper neuromuscular performance in male adult flies.** In contrast to the control (w[1118] and GFP RNAi) flies, the brain-specific knock-down of Hip14 causes a significant decline in neuromotor performance, especially in 28 days-old, aged males. However, this impaired climbing performance can be improved by the expression of constitutively active form of Rab2 small GTPase. The brain-specific loss of Patsas reduces the climbing performance, particularly in aged males, similar to Hip14. Black dots represent one measurement, circles indicate outliers, boxes represent the typical 50% of the climbed adults, the lines show the median and whiskers present the upper and lower quartiles, n = 20 flies per genotype. Significant differences are indicated by *** $p < 0.001$, ** $p < 0.01$, * $p < 0.05$, green and brown *-s represent significant difference compared to control (w[1118]) and control (GFP RNAi) respectively. Otherwise, significant differences between two genotypes are represented by black *-s over the clasps. (TIF)

**S1 Table. Raw quantification data for statistics represented on Fig panels.** (XLSX)

## Acknowledgments

We thank Ivett Répássy and Sarolta Pálfia for technical assistance, and colleagues and stock centers for providing reagents.

## Author contributions

**Conceptualization:** Győző Szenci, Gábor Juhász, Szabolcs Takats.

**Data curation:** Győző Szenci, Dorottya Károlyi, András Rubics, Zsombor Szőke, Gergő Falcsik.

**Formal analysis:** Győző Szenci, Gergő Falcsik, Tibor Kovács.

**Funding acquisition:** Attila Boda, Tibor Kovács, Péter Lőrincz, Gábor Juhász, Szabolcs Takats.

**Investigation:** Győző Szenci, Attila Boda, Anikó Nagy, Zsombor Szőke, Gergő Falcsik.

**Methodology:** Győző Szenci, Attila Boda, Anikó Nagy, Dorottya Károlyi, András Rubics, Zsombor Szőke, Gergő Falcsik, Tibor Kovács, Péter Lőrincz.

**Project administration:** Győző Szenci, Szabolcs Takats.

**Resources:** Győző Szenci, Gábor Juhász, Szabolcs Takats.

**Software:** Győző Szenci, Dorottya Károlyi, András Rubics.

**Supervision:** Győző Szenci, Tibor Kovács, Péter Lőrincz, Szabolcs Takats.

**Visualization:** Dorottya Károlyi.

**Writing – original draft:** Győző Szenci, Szabolcs Takats.

**Writing – review & editing:** Attila Boda, Anikó Nagy, Tibor Kovács, Péter Lőrincz, Gábor Juhász.

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
