## [Decision Letter · Decision Letter 0]

29 May 2025

PGENETICS-D-25-00145

Huntington’s disease-associated ankyrin repeat palmitoyl transferases are rate-limiting factors in lysosome formation and fusion

PLOS Genetics

Dear Dr. Takats,

Thank you for submitting your manuscript to PLOS Genetics. After careful consideration, we feel that it has merit but does not fully meet PLOS Genetics's publication criteria as it currently stands. Therefore, we invite you to submit a revised version of the manuscript that addresses the points raised during the review process.

Please submit your revised manuscript within 60 days Jul 28 2025 11:59PM. If you will need more time than this to complete your revisions, please reply to this message or contact the journal office at plosgenetics@plos.org. Please include the following items when submitting your revised manuscript:

We look forward to receiving your revised manuscript.

Kind regards,

Guang-Chao Chen

Academic Editor

PLOS Genetics

Fengwei Yu

Section Editor

PLOS Genetics

Aimée Dudley

Editor-in-Chief

PLOS Genetics

Anne Goriely

Editor-in-Chief

PLOS Genetics

**Additional Editor Comments :**

The manuscript has been reviewed by three experts in the field. While they find the study potentially interesting, they note that the current data lack sufficient mechanistic insight and have requested additional experimental evidence to support the conclusions. Please consider their comments carefully and address all points raised.

**Journal Requirements:**

3) Please upload a copy of Figure 8 which you refer to in your text on page 18. Or, if the figure is no longer to be included as part of the submission please remove all reference to it within the text.

4) We notice that your supplementary Figure is included in the manuscript file. Please remove it and upload it with the file type 'Supporting Information'. Please ensure that each Supporting Information file has a legend listed in the manuscript after the references list.

5) We note that your Data Availability Statement is currently as follows: "All data is shown in the manuscript or attached as Supplementary Information. Quantification of data is also shown." Please confirm at this time whether or not your submission contains all raw data required to replicate the results of your study. Authors must share the “minimal data set” for their submission. PLOS defines the minimal data set to consist of the data required to replicate all study findings reported in the article, as well as related metadata and methods (https://journals.plos.org/plosone/s/data-availability#loc-minimal-data-set-definition).

**Reviewers' comments:**

Reviewer's Responses to Questions

**Comments to the Authors:**

**Please note that one of the reviews is uploaded as an attachment.**

Reviewer #1: Győző Szenci et al. present novel and compelling evidence that the palmitoyl-transferases Patsas and Hip14 are key regulators of the interplay between lysosomes and the secretory pathway. The authors show that both enzymes support efficient crinophagic degradation by promoting fusion of residual glue granules with PI3P-positive endosomes, a step required for crinosome formation. Perturbing either HIP14 or Patsas impairs lysosomal maturation and fusion, leading to glue-granule accumulation in the Drosophila salivary gland and to climbing defects in adult flies. These findings suggest that Patsas and Hip14 act as rate-limiting factors in lysosome biogenesis/fusion and may protect against Huntington-disease-like phenotypes. However, the precise mechanism by which HIP14 and Patsas control lysosomal dynamics, especially in neurons, remains unclear. I therefore have several questions and suggestions, detailed below.

Major

1. Each gene is targeted with a single RNAi construct, which raises concerns about potential off-target effects or variable knockdown efficiency. Including an independent RNAi line would greatly strengthen the conclusion that the observed phenotypes are specific and reproducible.

2. Are HIP14 and Patsas localized and functional in the same way in neuronal tissue as in salivary-gland cells?

3. In Fig1., the nuclei seem to be imaged at different optical depths in the various genotypes, which could create an artifactual difference in appearance. Please show all samples at the same z-plane so that layer-dependent variation is excluded.

4. In Fig 1, the number and size of Sgs3-dsRed granules are altered in Hip14 and Patsas RNAi. It remains unclear whether these changes reflect increased biogenesis or impaired maturation. Additional quantification would help clarify this point.

5. In Fig2 B-C, Patsas RNAi appears to decrease, whereas HIP14 RNAi increases, Sgs3-dsRed granule number. Please clarify whether this divergent pattern reflects differences in knock-down efficiency or distinct roles of the two PATs; a double knock-down or epistasis test would be informative.

6. In Fig3., only vesicle size is reported. Including LysoTracker-positive vesicle counts (and mean intensity per cell) would reveal whether vesicle abundance, not just enlargement, is affected.

7. In Fig 6, several relevant control groups appear to be missing. First, the Rab2 constitutively active condition in HIP14 KD is tested, but the Rab2 dominant-negative or Rab2 WT only are not shown—this raises the question of whether granule phenotypes are driven by Rab2 activity level alone.

Second, all data are from overexpression conditions; testing Rab2 CA in a rab2 mutant background would clarify whether the effect is specific and sufficient.

Finally, in Fig 6A–D, Sgs3-RFP granule number changes are observed, but their biological significance (increased synthesis vs. blocked clearance) remains unclear and should be further discussed or quantified.

8. In Fig 6G., The red channel is severely smeared/over-saturated, obscuring structural detail.

9. In Fig 7, control group flies show reduced climbing ability at 14 d and 21 d, raising concerns about the assay design or interpretation. Please clarify the cause of the poor performance in the control group—whether it relates to the specific genotype used, handling conditions, or assay settings—and consider replacing the control genotype if necessary.

10. To confirm that behavioural rescue is lysosome-dependent, pharmacological inhibition (e.g., chloroquine or bafilomycin) could be applied to test whether the rescue is abolished.

11. In the adult brain, Changes in Cathepsin L puncta size alone are not sufficient to establish lysosomal dysfunction. Co-staining with LysoTracker and Lamp1, quantifying Cat L puncta number, and measuring Magic-Red Cat L activity would substantiate the link to lysosome function.

12. In Fig 7, the Patsas RNAi group and the Rab2 mutant-only group are not clearly examined in the adult brain. Please clarify whether these genotypes were tested, and if not, provide an explanation or include the relevant data to complete the comparison.

Minor

1. In Line 346, Figure 7D is not referenced in the text.

Reviewer #2: In the present study Szenci et al., the authors investigated the role of Golgi-resident ankyrin repeat-containing palmitoyl transferases (PATs), Hip14 and Patsas (Drosophila homolog of HIP14L), using Drosophila larval salivary gland. The study examined their roles in secretory and lysosomal membrane trafficking. The authos found that these PAT enzymes equally contribute to the proper maturation and crinophagic degradation of glue secretory granules by mediating their fusion with the endo-lysosomal compartment. They are also required for lysosomal acidification and biosynthetic transport of the lysosomal hydrolase CathL. Hip14 is also essential for proper lysosome formation and neuronal function in adult brains. Over-activation of lysosomal biosynthetic transport and lysosomal fusions by the expression of the constitutively active form of Rab2 could compensate for the lysosomal dysfunction caused by the loss of Patsas or Hip14 both in larval salivary glands and neurons. These findings are interesting, yet the results from the images can conclude only as much and a lot of claims in the paper are over-interpreted. A lot of terms such as “lysosome biogenesis” or “lysosome maturation” are carelessly used in the paper, which needs to be modified. For all Figures, statistical analyses are overly simple, a full and more complete analysis will help to dissect the difference better. Figure legends are poorly written. A lot of essential information were not included, misleading readers. I listed some of the major and minor concerns below:

1. Figure 1: when assessing the crinophagic flux, the statistical analysis seems rather simple. A full and more complete analysis will help to understand more in-depth on the efficiency of crinophagy. What about the number of puncta? The percentage of different type of puncta over the total number? Do PAT expression restore the defect?

2. Figure 1: what’s PP on the upper right corner? What’s the label S3R? No enlarged insets for A-C? Figure legends need to be more clear otherwise it’s hard to follow what the authors are trying to say – all legends.

3. Crinophagic degradation: is it specific to sgs3 or also applied to other glue proteins? Are there other proteins that can be used as examples to generalize the principle?

4. According to the information in the introduction (lines 87-93), it seems like the maturation of granules involves homotypic fusion, progressive acidification, events based on granules themselves and fusion with lysosomes. Thus, it might not be feasible to analyze the maturation of granules solely based on the fusion with lysosomes.

5. Figure 1&2: why is forming a ring around the granule indicating fusion? The authors seem to suggest so for both endosomes and lysoosmes.

6. Figure 3: the identity of the LTR-positive acidic organelles need to be verified. using other markers? LTR labels all acidic organelles including granules.

7. Figure 3: the size of LTR organelles does not seem to expand upon Hip overexpression, compared to control in the images. The representative images and the statistics do not correlate.

8. Figure 4: EM images show supersized granules – are them comparable with the ones in Figures 1-3?What does the scale bar indicate? Same or different in these figures? These information need to be present in the legends, otherwise it is rather hard to follow.

9. Figure 5: analyzing the sorting and transport of a lysosome hydrolase does not equal to analyze lysosome biogenesis. There is a huge difference. The authors use lysosome biogenesis but no size or area of the lysosomes were measured throughout the study. Either the term needs to be corrected or the authors need to perform more experiment to clarify whether the lysosome biogenesis is really affected by PATs.

10. Figure 5: The author stated that cathL-positive compartments are fragmented. This is bizarre. From the images, one might be able to say that the cathL-positive or the Lamp1-GFP-positive lysosomes are more “puncta-like”, but no evidence has been shown that whether it’s being fragmented. Lamp1-GFP-positive puncta also exhibit different structure, more puncta-like.

11. Figure 5: why is Hip14 overexpression causing different effect on cath-L and Lamp1-GFP?

12. Figure 6: what's the difference between LTR and LTdR? why using two different versions throughout the study?

13. Figure 7: I don’t see a decline in the climbing ability (Figure 7D) in Hip14-RNAi-expressing animals.

Reviewer #3: Post-translational modifications (PTMs) of proteins play important roles in regulating the subcellular localization, function, and stability of proteins, diversifying the spectrum of proteosome and involving in multiple biological processes. Palmitoylation is one of the PTMs, catalyzed by DHHC proteins in the ER and golgi apparatus. It has been suggested that accumulation of mutant Huntingtin proteins can reduce the activity of DHHCs like HIP14 and HIP14L (Patsas). However, how the events are triggered and the subsequent of these PATs loss of function in the pathogenesis of Huntingtin disease remains largely unclear. In this manuscript, the authors used the salivary gland (SG) of <drosophila> to explore the roles of HIP14 and HIP14L in the exocytosis and lysosomal degradation of glue granules, and later examined the phenotypes of HIP14/L in neurons to implicate the possible roles JIP14/L played in the pathogenesis of Huntingtin disease.

Although the authors found that HIP14/L may participate in the lysosomal degradation of un-secreted glue granules in the SG, directly or indirectly, and affect the neuronal functions, the manuscript is basically describing the phenotypes. No mechanism of the function of HIP14/L is attempted to be addressed at all. As the enzyme responsible for protein palmitoylation, how do HIP14/L function in this process? Which substrates are they palmitoylated? Do they function simultaneously in the secretion of glue granules and lysosomal components or they involve in the degradation of non-secreted glue granules alone? Do they function similarly in neurons as they do in the SG? Considering theses important unanswered questions, I would not recommend the publication of this manuscript, unless these questions are properly addressed.

Other concerns:

1. As HIP14 and HIP14L are homolog, are they equally required or differentially in the degradation of unsecreted glue granules? If they play redundant role in this process, why depleting one of them causes defects? If they function differentially, why similar defects are observed when they are knocked down?

2. The authors showed that both the degradation of un-secreted glue granules and secretion (exocytose of lysosomal components to the PM) are affected, how are the degradation of these glue granules processed?

3. in Figure 1, the authors indicated that the stage of development is prepupal stage, this is very confusing. The authors should indicate the defined time point of developmental stage, such as indicated in Figure 2, 2 hours before pupation.

4. Again in Figure 1, as they examined the SG at the same developmental stage and the same marker (Sgs3-dsRed) is used, why there are dramatic differences in the number of Sgs3-red granules between Fig. 1A-C and Fig. 1E-G?

5. for the forced secretion/transport of glue granules by ectopic expression of constitutively active Rab2, as Rab2CA promotes the secretion of glue granules and fusion of un-secreted granules with lysosome, why there are so many glue granules observed in Rab2CA-expressing cells? The authors also found that ectopic Rab2CA overrides the defects observed in HIP14/L knockdown cells, how this happens?

6. the transition from the SG to the neurons/brain is not supported by the same mechanism as assumed by the authors.</drosophila>

**Have all data underlying the figures and results presented in the manuscript been provided?**

Reviewer #1: Yes

Reviewer #2: Yes

Reviewer #3: Yes

PLOS authors have the option to publish the peer review history of their article (what does this mean? ). If published, this will include your full peer review and any attached files.

**Do you want your identity to be public for this peer review?** For information about this choice, including consent withdrawal, please see our Privacy Policy .

Reviewer #1: No

Reviewer #2: No

Reviewer #3: **Yes: ** Zhouhua Li

**Figure resubmission:**
---

## [Decision Letter · Decision Letter 1]

17 Dec 2025

Dear Dr Takats,

We are pleased to inform you that your manuscript entitled "Huntington’s disease-associated ankyrin repeat palmitoyl transferases are rate-limiting factors in lysosome formation and fusion" has been editorially accepted for publication in PLOS Genetics. Congratulations!

Yours sincerely,

Guang-Chao Chen

Academic Editor

PLOS Genetics

Fengwei Yu

Section Editor

PLOS Genetics

Aimée Dudley

Editor-in-Chief

PLOS Genetics

Anne Goriely

Editor-in-Chief

PLOS Genetics

BlueSky: @plos.bsky.social

Comments from the reviewers (if applicable):

Reviewer's Responses to Questions

**Comments to the Authors:**

Reviewer #1: The revised manuscript has shown significant improvement. I have no further quesitons.

Reviewer #2: The authors have addressed most of my previous concerns and I recommend for publication now.

Reviewer #3: I have no further comments for the revision. THe authors addressed most of the comments.

**Have all data underlying the figures and results presented in the manuscript been provided?**

Reviewer #1: Yes

Reviewer #2: Yes

Reviewer #3: None

PLOS authors have the option to publish the peer review history of their article (what does this mean? ). If published, this will include your full peer review and any attached files.

**Do you want your identity to be public for this peer review?** For information about this choice, including consent withdrawal, please see our Privacy Policy .

Reviewer #1: No

Reviewer #2: No

Reviewer #3: No

**Data Deposition**

http://datadryad.org/submit?journalID=pgenetics&manu=PGENETICS-D-25-00145R1

**Press Queries**

---

## [Editor Report · Acceptance letter]

PGENETICS-D-25-00145R1

Huntington’s disease-associated ankyrin repeat palmitoyl transferases are rate-limiting factors in lysosome formation and fusion

Dear Dr Takats,

We are pleased to inform you that your manuscript entitled "Huntington’s disease-associated ankyrin repeat palmitoyl transferases are rate-limiting factors in lysosome formation and fusion" has been formally accepted for publication in PLOS Genetics! Your manuscript is now with our production department and you will be notified of the publication date in due course.

With kind regards,

Anita Estes

PLOS Genetics

On behalf of:
